# Gene therapy with feline anti-Müllerian hormone analogs disrupts folliculogenesis and induces pregnancy loss in female domestic cats

William A. Stocker [1] ✉, Lauren Olenick[2], Shreya Maskey[1,3], Denise Skrombolas[2], Haitong Luan[1], Sophie G. Harrison[1], Matt Wilson[2], Anne Traas[2], Mark Heffernan[2], Samantha Busfield[2], Kelly L. Walton[3] & Craig A. Harrison [1] ✉

For female domestic cats, ovariohysterectomy is the only method of inducing permanent infertility. However, hundreds-of-millions of free-roaming cats globally highlight the necessity for alternative contraceptive approaches. One strategy involves a single injection of vector delivering a fertility-inhibiting protein for lifetime contraception. Recent studies in mice and cats have identified anti-Müllerian hormone as an excellent candidate for this type of contraception. Here, we leverage our recent characterization of the molecular mechanisms underlying human anti-Müllerian hormone synthesis and activity, to generate potent feline anti-Müllerian hormone analogs. Single intramuscular delivery of these analogs to female cats using an adeno-associated viral vector leads to a greater than 1000-fold increase in feline anti-Müllerian hormone levels, which are sustained for 9 months. High serum anti-Müllerian hormone is associated with abnormal estrus cyclicity, non-follicular ovarian cyst formation, and a progressive decline in antral follicle numbers, however, the few surviving large follicles continue to ovulate. Unlike previous studies, supraphysiologic levels of anti-Müllerian hormone do not block conception, although they are incompatible with the maintenance of pregnancy. Our findings highlight the complexity of the effects of anti-Müllerian hormone on ovarian physiology but confirm that this growth factor is a candidate for fertility control in free-roaming cats.

Anti-Müllerian hormone (AMH), also known as Müllerian inhibiting substance (MIS), is produced by granulosa cells of growing ovarian follicles[1,2]. Considered an autocrine/paracrine factor within the ovary, AMH impacts the growth and survival of follicles in a complex, possibly species-specific, manner. In mice, AMH reduces activation of primordial follicles and their transition to the primary follicle stage[3], and this effect is replicated in multiple species in vitro following administration of recombinant AMH to ovary explant or whole ovary cultures[4–7]. Collectively, these studies have suggested that AMH's main role within the ovary is to conserve the primordial follicle pool and prolong fertility[3]. However, a recent report by Zhou and colleagues[2] indicated that, at least in mice, AMH constantly induces primary follicle

[1]Department of Physiology, Monash Biomedicine Discovery Institute, Monash University, Clayton, VIC, Australia. [2]Scout Bio, 601 Walnut St, Philadelphia, PA, USA. [3]School of Biomedical Sciences, The University of Queensland, Brisbane, QLD, Australia. ✉e-mail: william.stocker@monash.edu; craig.harrison@monash.edu

atresia in the normal functioning ovary and that preventing the antral follicle pool from becoming too large is this growth factor's primary function. Interestingly, mice lacking AMH also display increased follicular atresia, highlighting the critical homeostatic role this hormone plays within the ovary[8]. In granulosa cells from multiple mammalian species, AMH has been shown to reduce the expression and/or activity of aromatase, the enzyme that converts androgens to estrogens[9–13]. Moreover, AMH reduces FSH receptor expression in human granulosa cells, meaning that follicles only undergo FSH-dependent cyclic recruitment and proceed towards ovulation when AMH levels decline[11–13]. Adding to this complexity, AMH actually stimulates growth during the culture of preantral follicles from rats and primates[13–15] and, at high levels, it promotes premature luteinization of antral follicles in human ovarian xenografts[16].

Despite the multifaceted and, at times, contradictory ovarian actions attributed to AMH, the overriding view is that this growth factor acts as a brake on mammalian folliculogenesis. As such, several groups have utilized pharmacological or adeno-associated-viral (AAV) administration of recombinant AMH in mice to slow primordial follicle activation and preserve fertility during chemotherapy[17,18], or to act as a contraceptive[3]. In terms of contraceptive effects, Kano et al.[3] showed that gene therapy with AMH reduced primary follicle numbers in mice by 40-65% and almost completely blocked the progression of these follicles through to the secondary stage. Not surprisingly, the lack of growing follicles in mice expressing high levels of AMH corresponded with a loss of fertility and the induction of a hypergonadotropic hypogonadic phenotype[3]. These findings suggested that a single viral vector injection of AMH could form the basis for a permanent, non-surgical means of contraception in companion and free-roaming animals (a strategy termed vectored contraception)[19,20]. Recently, Vansandt et al.[21] tested this concept by delivering a single intramuscular dose of an adeno-associated virus expressing feline AMH (fAMH) to domestic cats. Interestingly, supraphysiological levels of AMH had different effects on folliculogenesis in cats than mice, although the outcome (no pregnancies or live births) was the same. Specifically, overexpression of fAMH in cats did not appear to inhibit preantral or early antral follicle growth, as was observed in mice, as levels of serum inhibin B (a marker of the growing follicle pool) and fecal estradiol (an antral follicle marker) did not change. The levels of other ovarian hormones, including inhibin A and testosterone, as well as estrus cyclicity, were similarly unaffected by increased circulating levels of fAMH[21]. Rather, based on increased serum luteinizing hormone, decreased fecal progesterone, and reduced luteal phase frequency, the authors proposed that high levels of AMH inhibited breeding-induced ovulation, resulting in durable contraception in the cat[21].

Although the contraceptive effects of AMH gene therapy in mice and cats are exciting, the differing mechanisms of action are confusing. Recently, we characterized the molecular interactions that govern human AMH synthesis and activity[22]. In the course of that study, we: (1) enhanced processing of the AMH precursor 9-fold by incorporating an ideal proprotein convertase cleavage site ([448]RKKR[451]); (2) demonstrated that AMH_RKKR was more potent than wild-type human AMH; (3) generated a series of AMH variants with enhanced affinity for the type II receptor (AMHR2); and (4) identified residues within the wrist pre-helix of AMH that mediate type I (ALK2/ALK3) receptor binding[22]. Collectively, our results indicated that as few as three amino acid modifications across both the cleavage site and AMHR2 interface could significantly increase the production and activity of human AMH.

In this study, we envisage that the in vivo generation of increased amounts of mature (i.e., active) AMH with enhanced receptor affinity will be more efficacious than wild-type AMH in disrupting folliculogenesis and inducing a contraceptive effect in cats. Thus, we incorporate three specific mutations (A477K/Q478K/G561S) into feline AMH and demonstrate in vitro that these modifications enhance processing and activity. AAV delivery of these feline AMH variants results in

sustained secretion of high levels of processed and active AMH. Surprisingly, high serum AMH disrupts the latter stages of folliculogenesis but does not block ovulation. As such, most cats conceive during the breeding trial, but none of the cats within the AMH overexpression groups give birth.

## Results

### Mutagenesis enhances feline AMH processing and activity

Wild-type fAMH (Fig. 1a and Supplementary Fig. 1), which was overexpressed by Vansandt et al. to induce contraception in domestic cats[21], is poorly processed (Fig. 1c) and has low signaling activity (Fig. 1d). Based on our understanding of the synthesis, processing and activity of human AMH (Supplementary Fig. 1)[22–25], we modified the AMHR2 binding site (G561S) and/or the proprotein convertase cleavage site (A477K/Q478K) of feline AMH (Fig. 1a, b). Following expression of the resulting AMH variants (termed fAMH, fAMH_RKKR and fAMH_RKKR/G561S) in HEK293T cells, conditioned medium was analyzed by Western blot to assess the efficiency of precursor cleavage and the amount of mature AMH secreted (Fig. 1c). As observed previously for human and mouse orthologues[22], the fAMH precursor (75 kDa) was only partially (~10%) processed to its mature/active form (13 kDa) in vitro. The incorporation of an ideal proprotein convertase cleavage site (RKKR) in the fAMH precursor, however, enhanced processing and the accumulation of mature fAMH (Fig. 1c). Incorporation of a G561S point mutation in the fAMH_RKKR mature domain did not affect this enhanced processing efficiency.

To confirm that enhanced processing increased fAMH activity; HEK293T cells transfected with a SMAD1/5/9-responsive transcriptional reporter and AMH receptors, were treated with 1:50 dilutions of conditioned medium from cells expressing fAMH, fAMH_RKKR or fAMH_RKKR/G561S (Fig. 1d). In line with the amount of mature protein present in these preparations (Fig. 1c), fAMH_RKKR induced a luciferase response that was ~4-fold greater than that induced by fAMH. As seen with human AMH[22], incorporation of the AMHR2 activating mutation (G561S) into fAMH_RKKR led to a further 1.5-fold increase in activity. The enhanced processing and activity of fAMH_RKKR and fAMH_RKKR/G561S, relative to wild-type fAMH, suggested that these variants would be very efficacious contraceptive agents in female domestic cats. As such, fAMH_RKKR and fAMH_RKKR/G561S were packaged in an AAVrh91 vector for subsequent in vivo experiments.

### Gene therapy results in sustained secretion of feline AMH

Eighteen female cats of breeding age were randomly allocated to one of three groups ($n = 6$/group) and monitored for 14 days before receiving a single intramuscular dose of AAV-empty vector ($1 \times 10^{13}$ GC; control), AAV-fAMH_RKKR ($1 \times 10^{13}$ GC) or AAV-fAMH_RKKR/G561S ($1 \times 10^{13}$ GC) (Fig. 2a). Cats had daily wellbeing checks for the duration of the study. No injection site reactions were observed in any of the cats, nor were any serious adverse findings reported during routine monitoring or physical examinations.

Two weeks prior to AAV delivery, AMH serum levels in female cats ranged between 0.24 ng/ml and 5.74 ng/ml (Fig. 2b). Following AAV-fAMH_RKKR or AAV-fAMH_RKKR/G561S delivery, circulating AMH increased rapidly in 11-of-12 treated cats, reaching levels ~7000-fold greater than controls within three weeks (Fig. 2c–e) (NB: AMH levels did not increase in one cat in the AAV-fAMH_RKKR group). These high serum levels of fAMH were maintained throughout the 280-day study period, demonstrating the robust transgenic expression of this growth factor following AAVrh91 vector delivery to skeletal muscle.

Subsequently, we examined processing and activity of transgenic fAMH. Serum from cats in the fAMH_RKKR and fAMH_RKKR/G561S groups was collected 2 weeks prior (day 0) and 3- (day 35), 4- (day 42), 6- (day 56), 16- (day 126) or 32- (day 239) weeks after AAV delivery. Based on ELISA measurements (Fig. 2d, e), ~1 μl of serum was diluted and analyzed by Western blot. As expected, no AMH was detected in

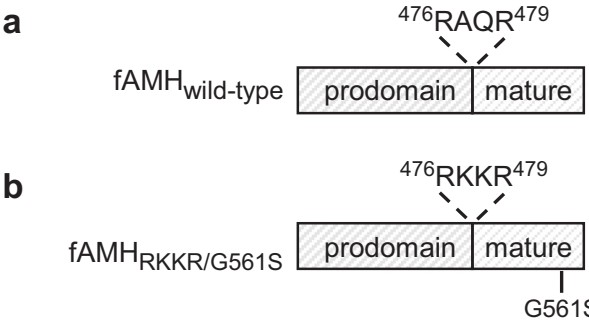

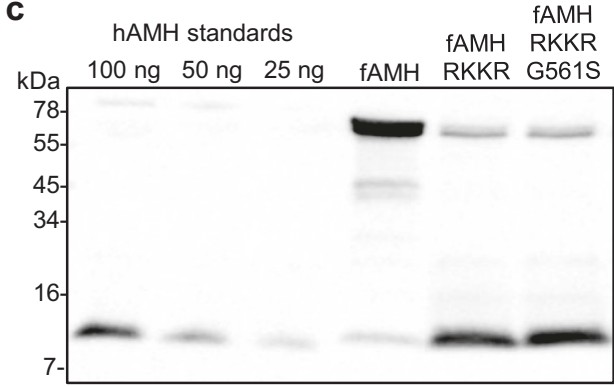

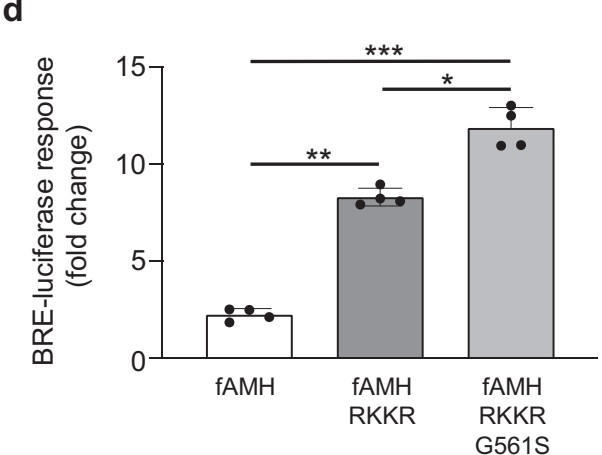

**Fig. 1 | Modifications in the pro- and mature domains of fAMH enhance processing and activity. a, b** The cleavage recognition motif (RAQR) in feline AMH was modified to the more efficient RKKR sequence. The G561S gain-of-function point mutation was then incorporated into the fAMH_RKKR construct to generate fAMH_RKKR/G561S. (**c**) To determine the effects on fAMH synthesis, conditioned medium from HEK293T cells transfected with either wild-type or mutant constructs was analyzed by reducing Western blot. Blots were probed with mAb-5/6, targeted to mature AMH. Recombinant human AMH was used as a positive control. Western blot shown in (**c**) is representative of three separate transfection experiments. (**d**) SMAD1/5/9-dependant luciferase reporter (BRE-luc) activity in AMH-responsive HEK293T cells following treatment with diluted conditioned medium containing fAMH variants. Luciferase activity is presented as the mean ± SD of quadruplicates ($n = 4$) from a representative experiment, relative to an adjusted value of 1.0 for the mean of wells which received fresh medium alone. The experiment was repeated 4 times. Data were analyzed using one-way ANOVA with Tukey's post hoc test (GraphPad Prism v.10). $P = 0.0146$ (*), $P = 0.0015$ (**) and $P = 0.0009$ (***). Source data are provided as a Source Data file.

the day 0 serum from any of the cats examined (Fig. 3a, lanes 3 and 8; and Supplementary Fig. 2), as control levels were well below the limit of detection for this assay (~0.5 ng/μl). In contrast, mature fAMH (13 kDa) was readily detected in the serum of cats 3-, 16- or 32-weeks after AAV-fAMH_RKKR delivery (Supplementary Fig. 2a and c), or 4-, 6-, 16- or 32-weeks after AAV-fAMH_RKKR/G561S delivery (Fig. 3a, lanes 4–6; and Supplementary Fig. 2b). Interestingly, no unprocessed fAMH precursor (75 kDa) was evident in these serum samples, suggesting that cleavage of fAMH_RKKR by proprotein convertases maybe even more efficient in vivo, than in vitro (compare with Fig. 1c).

To test circulating fAMH activity, serum from cats was collected 2 weeks prior (day 0) and at several intervals (either 3- (day 35), 4- (day 42), 5- (day 49), 16- (day 126) or 32- (day 239) weeks) after AAV delivery. Based on AMH ELISA measurements, serum collected post-AAV delivery was diluted 1:100 in fresh medium and used to treat HEK293T cells transfected with a SMAD1/5/9-responsive transcriptional reporter and AMH receptors. Day 0 serum from each of the cats failed to induce a transcriptional response, whereas diluted serum from days 35, 42, 49, 126, and 239 increased reporter activity ~6-fold (Fig. 3b and Supplementary Fig. 3a–c). In contrast, serum from cats that received AAV-empty vector showed no activity in this assay (Supplementary Fig. 3d). Together these results indicated that supraphysiological levels of fAMH_RKKR and fAMH_RKKR/G561S are continuously processed to their mature/active forms following transgenic expression in skeletal muscle.

### Prolonged supraphysiological AMH affects estradiol secretion

To assess the effect of high levels of active AMH analogs on the feline ovarian cycle, serum progesterone and estradiol levels were measured before (from day 0 to day 14) and after (from day 15-126) AAV delivery of AMH analogs or empty vector. Half the cats in the study displayed normal estrus cyclicity, as evidenced by a 2- to 3-fold increase in estradiol levels every 16-18 days, together with low and relatively stable levels of progesterone (Fig. 4a, b and Supplementary Fig. 4). The remainder of the cats experienced repeat spontaneous ovulations, a common response observed in group-housed cats[26], in which progesterone levels were elevated ~10-fold for a period of 40 days (estradiol levels remained at baseline during these progesterone excursions) (Fig. 4a, b and Supplementary Fig. 4). For cats with normal cycles (three controls, three fAMH_RKKR and three fAMH_RKKR/G561S) no differences were observed in estrus phase frequency across the first 90 days of hormone measurements. For the remaining cats (three controls, two fAMH_RKKR, and three fAMH_RKKR/G561S) the number of ovulatory cycles did not differ between groups. As a marker of ovarian steroid function, we measured serum levels of luteinizing hormone (LH), produced by the anterior pituitary, during estrus phases between days 0–90 of the study. LH levels were almost identical between control ($n = 5$ individual cats, $10.5 \pm 6.3$ ng/ml) and AMH overexpressing ($n = 17$ samples collected from 9 individual cats at different time points, $10 \pm 4.3$ ng/ml) cats, indicating that supraphysiological levels of AMH do not rapidly disrupt ovarian function (Supplementary Fig. 5a).

In the final 36 days of monitoring, however, potential changes in estradiol secretion and estrus cyclicity were observed in some cats with high circulating AMH, particularly those in the fAMH_RKKR/G561S group (Fig. 4b and Supplementary Fig. 4, arrowheads). In these cats, estradiol remained at, or near, baseline levels across this period, despite progesterone levels remaining low. In contrast, all control cats displayed normal estrus cyclicity across these final 36 days. These results suggested that after ~10 weeks exposure to high circulating levels of AMH, particularly the more potent analog, the population of antral follicles required to maintain estradiol production and estrus cycling was declining. An anticipated consequence of these declines would be reduced negative feedback by ovarian steroids to the hypothalamus and anterior pituitary, and a subsequent increase in gonadotropin (LH and FSH) release. Indeed, in serum samples collected at day 126 from AMH overexpressing cats, LH levels were

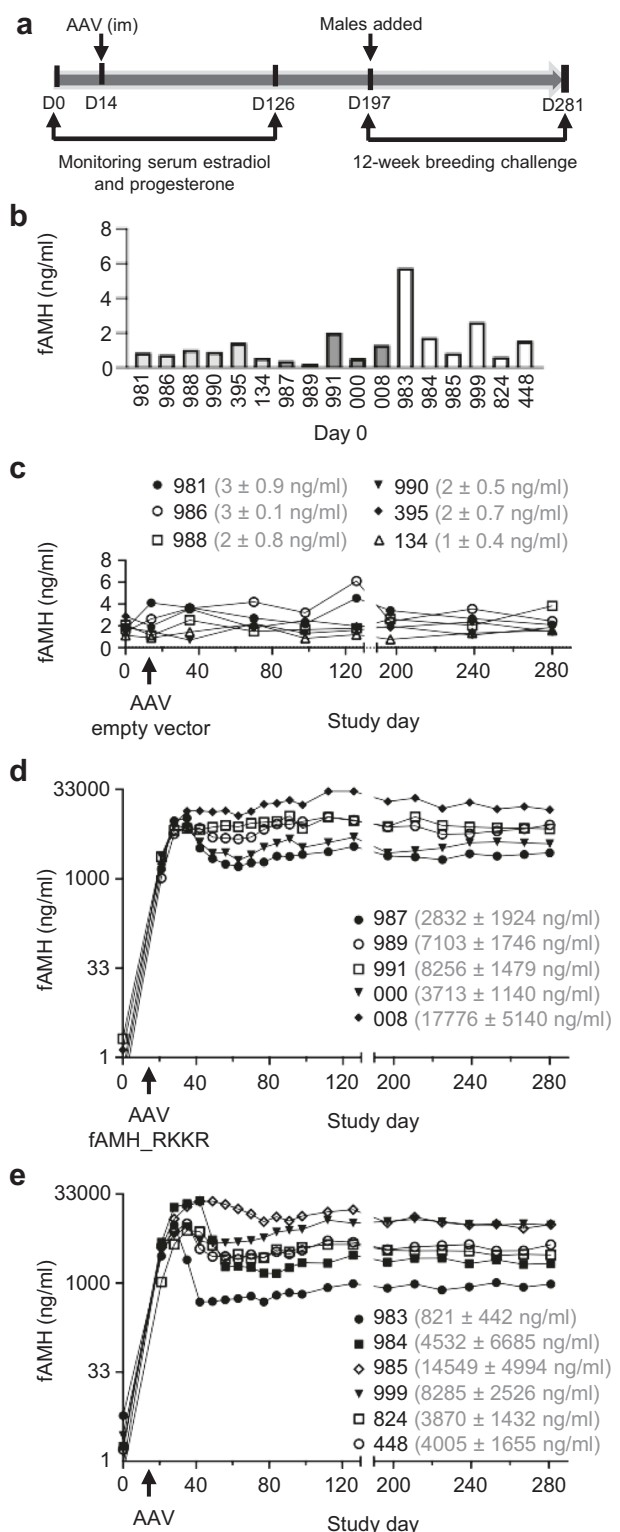

**Fig. 2 | Stable overexpression of fAMH analogs in healthy female cats. a** Study timeline: Eighteen sexually mature female cats, randomly separated into 3 groups, were monitored for 14 days before receiving intramuscular injections with $1 \times 10^{13}$ genome copies (0.5 ml) of AAV-empty vector ($n = 6$), AAV-fAMH_RKKR ($n = 6$) or AAV-fAMH_RKKR/G561S ($n = 6$). Serum estradiol and progesterone were measured every 2-to-3 days across the first 126 days of the study. Males were introduced at day 197 for a 12-week breeding challenge. **b** Serum AMH concentrations in all cats on day 0 of the study, as assessed by ELISA. Cats were ordered in control (light grey), fAMH_RKKR (dark grey) and fAMH_RKKR/G561S (white) groups and were identified by individual number. Serum AMH levels in cats before (0-14 days) and after (14-281 days) receiving an intramuscular injection of AAV-empty vector (**c**), AAV-fAMH_RKKR (**d**) or AAV-fAMH_RKKR/G561S (**e**). NB: One cat in the fAMH_RKKR group failed to respond. Mean ± SD serum AMH levels for each cat between days 35 and 281 are stated in brackets. Source data are provided as a Source Data file.

one of five intact males was introduced to each group. The male cats were rotated amongst the groups at least every 2 weeks over a 12-week period. Female cats were evaluated weekly for pregnancy via ultrasound and palpation starting two weeks after the introduction of the males. In the control group, all females ($n = 6$) conceived, with pregnancies detected 5-10 weeks after first contact with males. All but one of the control females gave birth to between 1 and 4 healthy kittens (Table 1). In the fAMH_RKKR and fAMH_RKKR/G561S groups, 3-of-5 and 5-of-6 females, respectively, conceived, with pregnancies detected 5-11 weeks after exposure to males. Interestingly, no pregnant females with high circulating levels of AMH analogs gave birth (Table 1). Transabdominal ultrasonography and physical examination suggested that pregnancy loss in these females was associated with fetal resorption, rather than abortion (refer to comments in Supplementary Table 1). For example, comments for female #984 from the fAMH_RKKR/G561S group indicated that pregnancy was detected via ultrasound four times between study days 239-260, but that: (1) the ultrasound was unclear on day 267, (2) the fetus appeared underdeveloped for length of pregnancy on day 274 and, finally, (3) nothing was detectable by ultrasound on day 281. Similarly, for female #989 from the fAMH_RKKR group, pregnancy was detected by ultrasound on day 232 and day 246, but the ultrasound was unclear on the intervening (day 239) and subsequent (days 253 and 260) assessments, and by day 274, this female was no longer deemed pregnant.

Based on a typical gestation period of 65 days in cats[27], we determined that pregnancies were first detected in females in this study at ~3 weeks gestation (control females gave birth 38–43 days after pregnancy detection). It was not possible to determine exactly when females with supraphysiological AMH levels lost their pregnancies, as some of these animals were only 4–5 weeks pregnant at the conclusion of the breeding trial, when ultrasounds stopped. However, based on the occurrence of unclear ultrasounds[28] in females who became pregnant early in the study, it is likely that high AMH levels disrupted the maintenance of pregnancy around 6-7 weeks of gestation (Supplementary Table 1).

### Folliculogenesis is disrupted in cats expressing high AMH

At the end of the study (i.e., after all kittens from the control group were weaned), a single ovary from each cat was removed under general anesthesia for histological assessment (Supplementary Fig. 6). As histology was performed 8 months after estrus cycle measurements ended (Fig. 4) and 3 months after the conclusion of the breeding study (Table 1), the observed morphology may not be entirely reflective of ovarian physiology during those periods. Nevertheless, the histology provides interesting insights into the physiological/pathological effects of AMH analogs on the feline ovary. Surprisingly, the number of primordial, primary/secondary and preantral follicles did not differ across the control, fAMH_RKKR and fAMH_RKKR/G561S groups (Fig. 5a). However, control cats had significantly ($P < 0.001$) greater numbers of small antral follicles in their ovaries, relative to cats with

significantly higher in cats with progressive cycle abnormalities ($n = 5$), compared to those with normal cycles ($n = 3$) ($21.5 \pm 3.1$ ng/ml vs. $7.7 \pm 4.7$ ng/ml, $P < 0.01$). (Supplementary Fig. 5b).

### Pregnancy in cats is disrupted by supraphysiological AMH

As the inhibitory effects of the feline AMH analogs on estradiol secretion and cycling appeared to be progressive, we waited an additional 10 weeks (until day 197) before initiating a 12-week breeding trial (Fig. 2a). The cats were randomly re-assigned to three new groups and

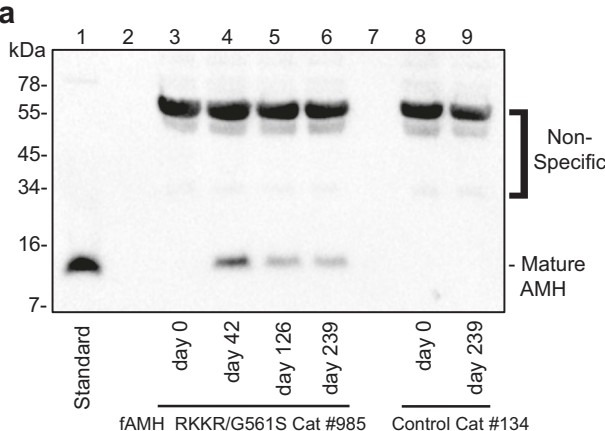

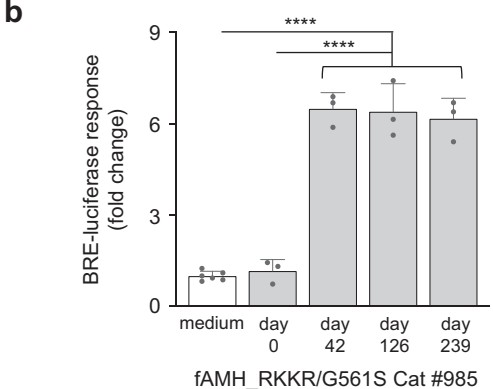

**Fig. 3 | In vivo processing and activity of transgenic fAMH. a** Based on ELISA measurements, -1 µl of serum collected at various times (day 0, day 42, day 126 and day 239) from cat #985, in the fAMH_RKKR/G561S group, was diluted and analyzed by Western blotting, with samples run under reducing conditions. Serum collected on days 0 and 239 from cat #134, in the control group, was run as a negative control. Recombinant mature human AMH was used as a positive control. Blot was probed with mAb-5/6, targeted to the AMH mature domain. **b** Serum from cat #985 in the fAMH_RKKR/G561S group was collected 2 weeks prior (day 0) and at different times (4- (day 42), 16- (day 126) or 32- (day 239) weeks) after AAV delivery. Based on AMH ELISA measurements, serum was diluted 1:100 in fresh medium and used to treat HEK293T cells transfected with a SMAD1/5/9-responsive transcriptional reporter (BRE-luc) and AMH receptors. Luciferase activity for cells treated with diluted cat serum is presented as the mean ± SD of triplicates ($n = 3$) from a representative experiment, relative to an adjusted value of 1.0 for the mean of the wells which received medium alone ($n = 6$). The experiment was repeated 4 times. Data were analyzed using one-way ANOVA with Tukey's post hoc test (GraphPad Prism v.10). $P < 0.0001$ (****). Source data are provided as a Source Data file.

high circulating levels of AMH. Large antral follicles also trended higher in control cats than in cats within the fAMH_RKKR and fAMH_RKKR/G561S groups, although this difference was not significant (Fig. 5a). Atretic follicle numbers were not significantly different between groups. Interestingly, despite the apparent lack of antral follicles, corpora lutea (CL) were present within the ovaries of most cats with high serum AMH levels, and numbers of CL were not significantly different from those observed in control cats (Fig. 5a).

In terms of pathological alterations, a non-follicular ovarian cyst was found in 1-of-6 (16%) control cats (#395, Fig. 5b), which is in line with a recent study examining the presence of ovarian cysts in clinically healthy cats[29]. In contrast, non-follicular ovarian cysts were found in 40% of fAMH_RKKR cats and 50% of fAMH_RKKR/G561S cats, respectively (Fig. 5b). These large cysts, which were typically accompanied by lymphatic dilation, caused significant compression of the ovarian parenchyma. In a separate cat within the fAMH_RKKR/G561S

group (#984), a cystic corpus luteum, which compressed the adjacent ovarian cortex, was also observed. Finally, of the five cats across the AMH groups that had no detectable ovarian cysts, three had numerous regressing corpora lutea, a finding not observed in any of the control cats (Fig. 5b).

Perhaps reflecting the transitory nature of the above ovarian changes, no correlation could be made between CL numbers and/or ovarian cyst formation at the end of the study and which cats conceived during the breeding trial. Together our findings indicate that high levels of active AMH have multi-faceted effects on the feline ovary, which either directly or indirectly, block the maintenance of pregnancy.

## Discussion

In Australia, depending on environmental conditions, the feral cat population can be as large as the pet cat population (5.3 million) and these cats kill 2 billion mammals, birds and reptiles every year[30]. Cats have caused profound species loss in Australia and are a recognized threat to 200 nationally listed threatened species, including the Greater Bilby and Numbat[30]. In response, the Australian government has recently released a draft threat abatement plan for predation by feral cats[31]. Part of this plan revolves around new and improved fertility control measures to reduce the number of pet cats joining the feral population. One approach that could meet this criterion is vectored contraception, which involves a single administration of gene therapy that drives the expression of a fertility-inhibiting protein for lifetime contraception[19]. Recent studies in mice and cats suggest that AMH is an excellent candidate for vectored contraception[3,21].

Based on our characterization of the molecular interactions that underlie the synthesis of human AMH[22] and the AMH:AMHR2 crystal structure[24], we generated feline AMH analogs with (1) enhanced processing to the mature/active form (fAMH_RKKR), or (2) enhanced processing and increased potency (fAMH_RKKR/G651S). It was anticipated that these feline AMH analogs would be more efficacious than the wild-type protein in inducing contraception in the female domestic cat[21]. Following gene therapy, fAMH_RKKR and fAMH_RKKR/G561S were fully processed to produce supraphysiological levels of active AMH in cats, which were sustained for the 9-months of this study. The serum levels of AMH analogs in treated cats ranged from 959 ng/ml to 14,528 ng/ml on day 280 of the study, which was 433-6573 times the AMH levels in control cats. Importantly, the levels of fAMH_RKKR and fAMH_RKKR/G561S were equivalent to the contraceptive-inducing levels of wild-type fAMH observed in the study by Vansandt and colleagues[21].

In the mouse, transgenic human AMH acts as a contraceptive by reducing primordial follicle activation and inducing granulosa cell quiescence, which blocks the development/survival of early preantral follicles[3,6]. Feline AMH overexpression in mice has similar effects, inducing a 60% reduction in primary follicle numbers and an almost complete loss of secondary and antral follicles[21]. In cats, however, we found that high levels of fAMH have no effect on the numbers of primordial, primary, or secondary ovarian follicles. An interpretation of these results is that small follicles in the feline ovary do not express the signaling receptors and/or downstream transcriptional machinery required to be growth-inhibited by AMH, even at supraphysiological levels. To date, the only study to have assessed AMHR2 expression in the feline ovary did not report immunopositivity in primary or secondary follicles[32].

Sex steroid levels were measured across the first 126 days of our study, with adeno-associated viral delivery of AMH or empty vector occurring on day 14. Cats in the three groups either displayed normal estrus cycles (9 of 17) or normal ovulatory cycles (8 of 17) across the first 3 months of measurements, indicating that high serum AMH did not rapidly affect the growth of antral follicles (the source of cyclic E2) or the ovulatory process. These findings did not support the previously

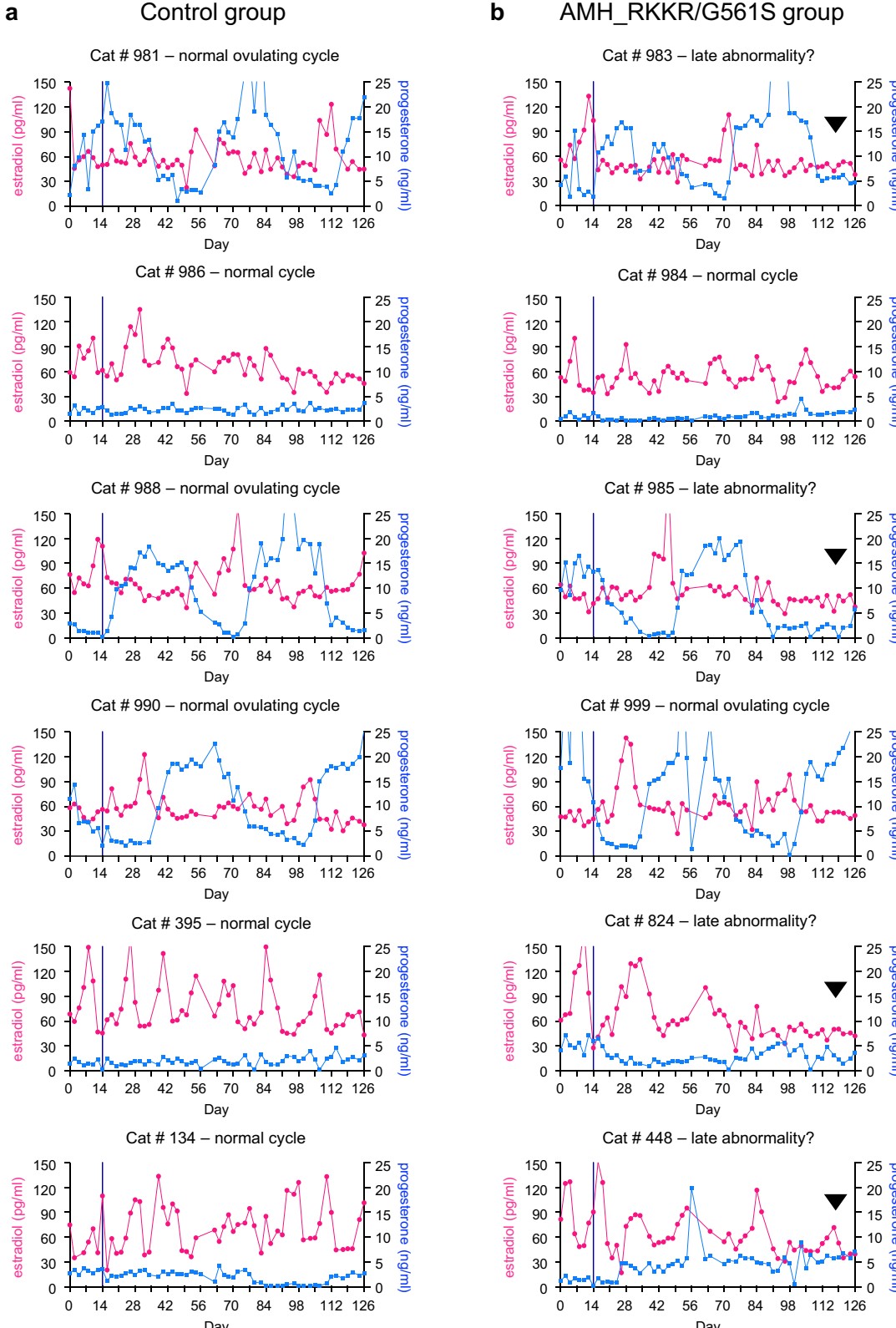

**Fig. 4 | Effect of elevated serum levels of fAMH_RKKR/G561S on estrus cycles and steroid hormone levels.** Serum steroid hormone levels in cats ($n = 6$) injected with AAV-empty vector (**a**) or cats ($n = 6$) injected with AAV-fAMH_RKKR/G561S (**b**) between days 0 and 126 of the study. Cyclic increases in estradiol levels (pink line) indicated estrus phases, while prolonged increases in progesterone levels (blue line) indicated luteal phases, following spontaneous ovulation. Potential abnormalities in estrus cyclicity in some cats in the AAV-fAMH_RKKR/G561S are indicated (**b**, arrowhead). The day of AAV delivery (day 14) is indicated (vertical dark blue line). Source data are provided as a Source Data file.

**Table 1 | Pregnancy and live birth outcomes**

| Cohort | Number of cats with detected pregnancies | Number of Litters | Kittens born alive | Stillborn |
|---|---|---|---|---|
| Control (n = 6) | 6 | 5 | 12 | 1 |
| fAMH_RKKR (n = 5) | 3 | 0 | 0 | 0 |
| fAMH_RKKR/G561S (n = 6) | 5 | 0 | 0 | 0 |

Source data are provided as a Source Data file.

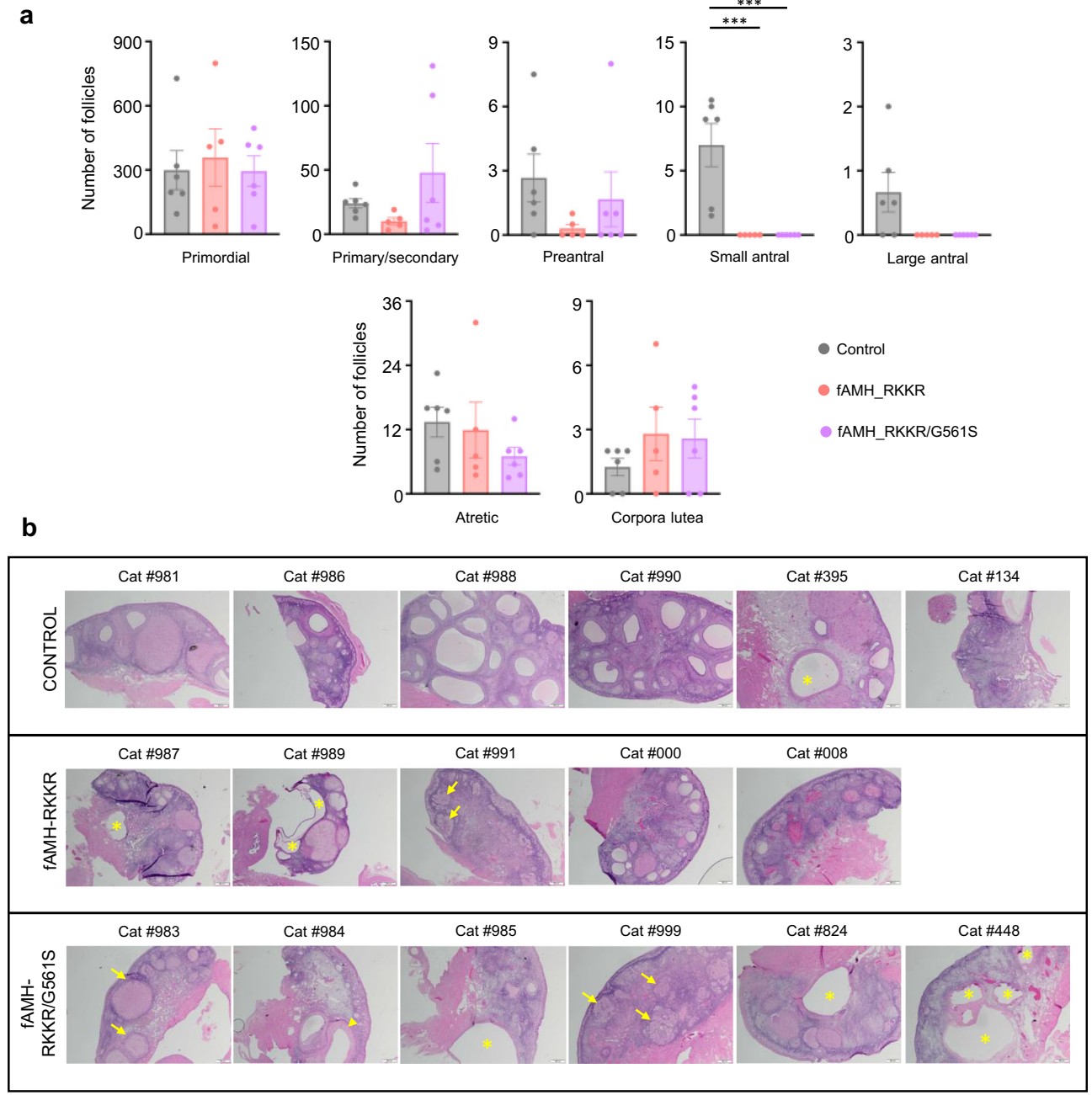

**Fig. 5 | Analysis of ovarian function in cats overexpressing fAMH analogs.** Ovaries (n = 6, 5 and 6: for the control, fAMH_RKKR and fAMH_RKKR/G561S groups, respectively) were sectioned at 3-5 μm before being stained with hematoxylin and eosin for histological analysis. **a** Primordial, primary/secondary, preantral, small antral, large antral and atretic follicles, together with corpora lutea, were counted for the control (grey), fAMH_RKKR (salmon), and fAMH_RKKR/G561S (purple) groups. Bars represent mean ± SE of the mean. Data were analyzed using one-way ANOVA with Tukey's post hoc test (GraphPad Prism v.10). (***) control vs. fAMH_RKKR: P = 0.0009; (***) control vs. fAMH_RKKR/G561S: P = 0.0006. **b** Representative ovarian sections from each of the cats in the study. Non-follicular ovarian cysts (asterisk), cystic corpora lutea (arrowhead), and regressing corpora lutea (arrows) are indicated. Source data are provided as a Source Data file.

proposed contraceptive mechanism of supraphysiological AMH in cats, i.e., that follicles reaching the antral stage cannot complete maturation and ovulate in response to an LH surge[21]. In the final month of hormone measurements, however, normal cyclic increases in estradiol secretion were blunted in some cats with high serum AMH (5-of-11), suggesting a gradual decline in antral follicle numbers. Histology, performed 8 months after the final hormone measurements, supports a concept of progressive ovarian dysfunction in the presence of high serum AMH, as antral follicles were undetected in all treated cats by this time (even those that were cycling normally at the end of hormone measurements). AMH has been shown to reduce FSH receptor expression in human granulosa cells[11] and, thereby, diminish FSH-dependent growth and selection of small antral follicles[12]. AMH similarly desensitizes growing mouse and macaque follicles to the effects of FSH[13,33]. These actions may underlie the progressive decline of antral follicles in the feline ovary in the current study.

Despite the disruption of estrus cyclicity and greatly reduced numbers of antral follicles, ovulation still occurred in most cats with high circulating levels of AMH, as evidenced by the presence of ovarian corpora lutea (CL) at the study endpoint. Although some of these CLs were clearly regressing, others appeared normal and likely arose from recent ovulations. Oocytes ovulated were fertilizable, as 8-of-11 cats overexpressing AMH had detectable pregnancies 5-11 weeks after the introduction of males. However, no pregnancy was maintained in cats with high serum AMH levels, with ultrasound recordings suggestive of mid/late-gestation fetal resorption. Importantly, endocrine abnormalities are often associated with pregnancy loss in cats[34].

Collectively, the phenotype observed in cats with supraphysiologic fAMH bears remarkable similarity to that reported for a mouse line with neuronal overexpression of hAMH (*Thy1.2-AMH*)[35]. *Thy1.2-AMH*[Tg] mice secrete relatively low levels of hAMH (1 nmol/L), however, females display large declines in follicles surviving beyond the primary stage[36]. Despite far fewer preantral/small antral follicles, *Thy1.2-AMH*[Tg] females cycle normally and have typical numbers of preovulatory follicles. Strikingly, although *Thy1.2-AMH*[Tg] females ovulate, conceive at normal rates, and carry normal numbers of midterm fetuses, they experience near-complete fetal resorption (miscarriage) by E13.5[35–37]. After extensive analysis, the authors concluded that a failure to downregulate luteinizing hormone (LH) secretion in early pregnancy in *Thy1.2-AMH*[Tg] dams was the most likely cause of fetal resorption. In support, mice overexpressing transgenic LH display an identical miscarriage phenotype[37].

As transgenic AMH has been promoted as a potential contraceptive in cats, we examined ovaries for any pathologic impact on folliculogenesis. A non-follicular ovarian cyst was observed in 1-of-6 cats within the control group (16%), however, 55% of cats with high serum AMH had large ovarian cysts. Most of these cysts appeared to originate in the rete ovarii, an appendage of the ovary, which is receiving renewed attention as a potential regulator of ovarian development and homeostasis[38]. Cystic rete ovarii occur occasionally in cats[29,39,40], and are generally considered incidental findings without clinical relevance. The rete ovarii cysts observed in cats with supraphysiologic AMH typically compressed the ovarian parenchyma, a finding that has been linked with secondary ovarian functional loss in other species[41].

Ours is the second recent study to use AMH gene therapy to prevent female cats from giving birth[21]. However, the proposed sequence of events induced by AMH to block live births differs between the studies. Vansandt and colleagues proposed that high AMH levels stall antral follicle maturation and inhibit LH-induced ovulation and, thereby, provide contraception in cats (indeed, none of the 6 cats that received AMH gene therapy in this original study conceived across two breeding trials)[21]. In contrast, the current study, which benefited from access to ovarian tissue, indicated that overexpression of AMH reduced the antral follicle pool without inhibiting ovulation/conception. The lack of live births was due to mid/late-gestation fetal resorption, rather than contraception.

Differences in AMH forms used (better processed, more active, in the current study) and the timing/number of breeding trials could partially explain the distinct findings of these studies. However, further research is clearly required to unravel AMH's potential therapeutic or pathologic impact on the feline ovary. Any future study should consider initiating earlier and more frequent breeding trials after AAV:fAMH delivery (e.g., 2–6, 8–12, 14–18 and 20–24 months) to determine how quickly elevated AMH disrupts the maintenance of pregnancy and, potentially, attains contraceptive efficacy. Sufficient serum should also be collected through the pre- and post-treatment periods, including during the breeding trials, to determine how hypothalamic-pituitary-ovarian function is disrupted and, thereby, contributes to the observed pregnancy failure. Although difficult, given the cost of feline studies, performing histology, immunohistochemistry and RNA sequencing on ovaries at different timepoints after AMH overexpression (e.g., 4, 8 and 12 months) would provide invaluable information on the progressive decline in ovarian function and the formation of ovarian cysts. In pregnant mice, disruption of hypothalamic-pituitary function due to, and together with, elevated AMH alters placental steroid metabolism, with smaller litter sizes observed in these mice due to an increased number of aborted embryos[42]. Future cat studies should therefore assess placental function, to determine whether this is also a contributing factor to the pregnancy loss we observed in cats following AMH overexpression.

## Methods

### Animals

The study protocol (21SCB021) was assessed by LFM Quality Laboratories, Inc. (USA) internal Institutional Animal Care and Use Committee (IACUC), and complies with all relevant ethical regulations.

Domestic cats (*Felis catus*) were sourced from LFM Quality Laboratories, and were all in good physical health as determined by physical examinations performed during the acclimation phase. Female cats (*n* = 18) were between 19.5 months and 51 months old at the start of the study. Male cats (*n* = 5) were 13 months old when introduced into the study. Animals were group housed with free roaming throughout the study. Rooms were temperature and humidity controlled and under a 14 h/10 h light-dark cycle. Fresh feed was provided once daily, and water offered *ad libitum*. Environmental enrichment was supplied to animals in compliance with the internal enrichment program.

### Generation of feline AMH variant expression constructs
The amino acid sequence of feline AMH (GenBank ID: XP_011286375.2) was codon-optimized and synthesized using Invitrogen GeneArt Gene Synthesis Services, followed by cloning into the mammalian expression vector pcDNA3.1(+), between the *Nhe*I and *Xho*I sites (Thermo Fisher Scientific, Waltham, MA, United States). The feline AMH amino acid sequence was aligned with the human AMH sequence (GenBank ID: AAH49194.1) using Clustal Omega (Conway Institute, University College Dublin, Dublin, Ireland; accessible at: http://www.clustal.org/omega/). Mutagenesis of the feline AMH expression construct was performed using the QuikChange Lightning Site-Directed Mutagenesis Kit (Agilent Technologies, Santa Clara, CA, United States) and custom mutagenic primers (Sigma-Aldrich, St. Louis, MO, United States). The sense and anti-sense primer nucleotide sequences used to incorporate the A477K/Q478K mutations were 5'-gccggaccagctagaaagaagagaagtgctggtgc-3' and 5'-gcaccagcacttctcttctttctagctggtccggc-3', respectively. The sense and anti-sense primer nucleotide sequences used to incorporate the G561S mutation were 5'-gcctacagcctacgccagcaagctgctgatctc-3' and 5'-gagatcagcagcttgctggcgtaggctgtaggc-3', respectively.

### Transient expression of feline AMH variants in HEK293T cells

For the production of recombinant feline AMH variants in vitro, HEK293T cells (Merck KGaA, Darmstadt, Germany; catalog # 12022001-1VL) were plated at $4 \times 10^5$ cells/well in 12-well plates in Dulbecco's modified Eagle medium (DMEM) supplemented with 10% fetal calf serum (FCS) and incubated at 37 °C in 5% $CO_2$. After overnight incubation, plasmid DNA (2.5 μg/well) was combined with poly-ethylenimine (PEI)-MAX (Polysciences, Warrington, PA, United States) and after 30 minutes DNA-PEI complexes were added directly to cells and incubated in OPTI-MEM (Life Technologies, Carlsbad, CA, United States) medium for 4 hours, before replacing with fresh OPTI-MEM and incubating a further 90 hours before collection of conditioned medium containing the secreted feline AMH variants. The conditioned medium was then centrifuged, and the supernatants stored at −20°C.

### Assessing processing and activity of feline AMH variants in vitro

To assess the in vitro biosynthesis of feline AMH variants, conditioned medium collected from transfected cells (described above) was thawed and concentrated 12.5-fold using Nanosep microconcentrators (3 kDa MW cut-off; Pall Life Sciences, Port Washington, NY, United States) before separation of reduced samples by 10% SDS-PAGE and Western blot. The primary antibody (mAb-5/6), targeted to a region near the C-terminus of AMH (used at a 1:5000 dilution), was from Abcam (Cambridge, UK; catalog # ab24542). The secondary antibody (diluted 1:10,000) was Amersham ECL sheep-derived horseradish peroxidase-conjugated anti-mouse IgG (Cytiva, Mount Waverly, VIC, Australia; catalog # NA931-1ML), with detection of immunoreactive proteins using Lumi-light chemiluminescence reagents (Roche, Basel, Switzerland) and a ChemiDoc™ MP system (Bio-Rad, Hercules, CA, United States) with Image Lab™ software (version 4.1; Bio-Rad). Recombinant human AMH mature protein (R&D Systems, Minneapolis, MN, United States) was used as a positive control and as standards of known mass. All blots presented in either the main figures or Supplementary Information are shown in an uncropped and unprocessed form in the source data file.

To confirm the activity of the feline AMH variants produced in vitro, a cell-based luciferase reporter assay was utilized[22,24]. In brief, HEK293T cells were seeded in 96-well plates at $1.5 \times 10^4$ cells/well. After overnight incubation, the cells were transiently transfected using Lipofectamine 2000 (Life Technologies) with a mixture of plasmid DNA (100 ng/well), consisting of: a SMAD1/5/9-responsive transcriptional reporter (4xBRE-luc, 98.9 ng) and AMH receptors (AMHR2 and ALK2, 0.8 ng and 0.3 ng, respectively). After an additional 24 hours, conditioned medium from HEK293T cells transiently expressing feline AMH variants (described above) was diluted out equally in fresh serum-free medium (DMEM supplemented with 1 mM sodium pyruvate and 0.01% BSA), and the dilutions were used to treat the AMH-responsive HEK293T cells overnight. The medium was then removed, and the cells were lysed in solubilization buffer (26 mM glycylglycine (pH 7.8), 16 mM $MgSO_4$, 4 mM EGTA, 900 μM dithiothreitol, 1% Triton X-100). The lysate was transferred to a white 96-well plate and luminescence was measured immediately following the addition of the substrate luciferin (Promega, Madison, WI, United States), using a CLARIOstar microplate reader (BMG Labtech, Ortenberg, Germany). Raw luciferase data was exported to Microsoft Excel (version 2019; Microsoft Corporation, Redmond, WA, United States) for initial analysis to determine the relative change in activity over baseline for each treatment. Fold change data was then transferred to GraphPad Prism (version 10; GraphPad Software, Inc. Boston, MA, United States) for graphing and statistical analysis.

### AAV vector production

The University of Pennsylvania Gene Therapy Program's Vector Core produced recombinant AAV vectors[43–45]. Briefly, to generate AAV vectors, codon-optimized transgenes encoding each of the fAMH analogs were cloned into an expression vector containing a hybrid cytomegalovirus enhancer/chicken beta-actin promoter. The expression constructs were flanked by 5′ and 3′ AAV inverted terminal repeats (ITR) and were packaged in an AAV Clade A rh91 capsid (AAVrh91) vector[46] by triple transfection with helper and trans plasmids. The AAV vector particles were subsequently purified by iodixanol gradient purification before being titred by Taqman quantitative PCR.

### Adeno-associated vector delivery of fAMH analogs to cats

An expression and breeding study in cats was performed at LFM Quality Laboratories, Inc. (USA), following institutional ethics review panel approval. Eighteen female cats of breeding age were randomly allocated to one of three groups ($n = 6$/group). The cats were monitored for 14 days prior to administration of the AAV-fAMH vectors. A single intramuscular administration of $1 \times 10^{13}$ genome copies (GC, 0.5 ml) of AAV-empty vector, AAV-fAMH_RKKR or AAV-fAMH_RKKR/G561S was given on day 14. General health observations were performed at least once daily throughout the study and injection site observations were performed post-dose at 2–4 h and at days 1-3.

### Serum collection and hormone measurements

1–2 ml of blood was collected without anticoagulant from the jugular or cephalic vein and allowed to clot prior to centrifugation at $1000 \times g$ for 10 min at 2–8 °C. Serum was then used for hormone and transgene expression analysis. Serum progesterone and estradiol were measured every second to third day, from day 0 through to day 126 at Antech laboratories. Expression of the fAMH analogs in the serum was measured weekly using a canine/feline-AMH ELISA (AL-116, Ansh Labs, USA)[47]. Analyses were performed concurrently in duplicate. Briefly, duplicate serum or standards (25 μl/well) were added to AMH antibody-coated micro-titer wells and incubated with 75 μl of AMH assay buffer for 1 h at room temperature. After washing 5 times, 100 μl of the antibody-biotin conjugate RTU was added to each well and incubated for 30 min at room temperature. After washing 5 times, 100 μl of streptavidin horseradish peroxidase conjugate solution was added to each well and incubated for 30 min at room temperature. After washing 5 times, 100 μl of TMB chromogen solution was added in each well and incubated for 12 min at room temperature. After 100 μl of stopping solution was added in each well, the absorbance of color in the microplate was evaluated in a microplate reader at 450 nm. Serum LH concentrations were determined from concurrent duplicate measurements with a feline LH ELISA kit (Catalog # MBS284581, MyBio-Source, San Diego, CA, USA), used according to the manufacturer's instructions.

### Assessing processing and activity of feline AMH analogs in vivo

To assess the processing efficiency of circulating feline AMH variants following transgenic expression in skeletal muscle, ~1 μl of serum (collected both before and at different time points after AAV delivery) was diluted in NuPAGE™ LDS Sample Buffer (Life Technologies) containing 2-mercaptoethanol (Bio-Rad) and analyzed by Western blot, as described above. To confirm the feline AMH variants circulating in vivo following AAV delivery were active, we used a HEK293T cell-based luciferase reporter assay, as described above. Briefly, cat serum samples (collected both before and at different time points after AAV delivery) were diluted 1:100 in fresh serum-free medium. The dilutions were then used to treat AMH-responsive HEK293T cells overnight, after which luciferase activity was measured.

### Breeding trial

On day 197, the cats were randomly re-assigned to three new groups and one of five intact males was introduced to each group. The male cats were rotated amongst the groups at least every 2 weeks over a 12-week period. Female cats were evaluated weekly for pregnancy via ultrasound and palpation starting two weeks after the introduction of

the males. A final pregnancy assessment was performed 5-6 weeks after the removal of the male cats.

## Histology

At the end of the study, a single ovary from each cat was removed under general anesthesia. Ovaries were only collected after any kittens were weaned. Seventeen feline ovarian specimens were submitted for histologic evaluation. The specimens were blinded in terms of treatment and control groups. When possible, the ovarian tissue was bisected to provide two sections of the same ovary on one slide. Tissues were processed routinely to glass slides that included approximately 3-to-5-micron sectioning and staining with hematoxylin and eosin. Slides were cover-slipped and submitted for microscopic evaluation by a board-certified veterinary pathologist/theriogenologist. Follicles were quantified by light microscopy.

The following criteria were utilized to quantify follicular structures[48,49]: (1) Primordial follicles - an oocyte 20 to 30 μm in diameter, surrounded by a single layer of flattened follicular cells; (2) Primary/Secondary follicles - an oocyte 30 to 75 μm in diameter with an obvious zona pellucida, surrounded by a single layer or multiple layers of cuboidal cells, and a total follicle diameter of 100 to 400 μm, depending on the number of granulosa cell layers, but with no evidence of space formation between the granulosa cells. The outermost layer of granulosa cells are encapsulated within a basement membrane, which by the later stages will also have a theca cell layer on the outside; (3) Preantral follicle - an oocyte surrounded by multiple layers of granulosa cells and fluid-filled spaces have started to form in between the granulosa cells, but a fully defined follicular antrum is not present. A theca cell layer is present; (4) Antral follicle - has a fully formed fluid-filled follicular antrum. Oocyte diameter is typically 75 to 100 μm and is surrounded by a corona radiata and cumulus oophorus. Two or three layers of theca cells are present in small antral follicles, with large antral follicles containing more layers. Antral follicles <2 mm and >2 mm in diameter were quantified separately. Corpora lutea, atretic follicles at any stage, and remnant zona pellucida were also quantified. Morphologically, corpora lutea are large structures filled with cells and can have visible lipid deposits. Atretic follicles were characterized by a degenerating oocyte together with zona pellucida, or remnants of the zona pellucida where the oocyte was no longer present. The granulosa cell layers become disorganized, and depending on the follicle stage when atresia begins, a shrinking/collapsing follicular antrum may also be visible, with granulosa cells present within the antrum. Ovarian cysts were classified based on the following criteria[29]: a fluid-filled structure either on, in, or next to the ovary, and greater than 3.5 mm in diameter. Most cysts likely originated in the rete ovarii[39], and were characterized as a fluid-filled structure lined by a layer of cells (flattened, cuboidal or columnar, with or without cilia), and held intact by a narrow layer of connective tissue. A cystic corpus luteum in one cat was characterized by a fluid-filled structure surrounded by luteinized cells.

## Statistics

Activity of feline AMH variants. Conditioned medium from HEK293T cells transfected with fAMH constructs; or serum from cats injected intramuscularly with AAV-empty vector, AAV-fAMH_RKKR or AAV-fAMH_RKKR/G561S, were tested for their ability to activate a SMAD1/5/9-responsive luciferase reporter in HEK293T cells. Luciferase activity is presented as the mean ± SD of quadruplicates or triplicates, respectively, from a representative experiment, relative to an adjusted value of 1.0 for the mean of wells that received medium alone. Each assay was repeated 4 times. Data were analyzed using one-way ANOVA with Tukey's post hoc test (GraphPad Prism v.10). Significance was shown when $P < 0.05$.

Sample size, randomization, and blinding. Prior studies examining vectored AMH in mice[3] showed complete infertility and revealed large differences in the number of growing primary, secondary, and antral follicles, as well as corpora lutea within the ovaries. Follow-up studies in cats[21] led by the same team indicated that vectored AMH delivery in cats also resulted in complete infertility. As such, it was predicted that mean follicle counts and fertility parameters would be vastly different between experimental groups, consequently requiring less animals to reach statistical significance. Based on these studies, we performed power calculations assuming a minimum mean difference between follicle counts of ≥ 50% (with standard deviations of ≥ 25%), with a probability of $P < 0.05$ (alpha level 5%), and power of 80% (beta level 20%). Using these values, we estimated that a minimum of $n = 4$ cats would be sufficient to reach statistical significance in follicle counts between groups. For fertility measures, we predicted that the AMH vectored contraceptive would induce complete infertility, therefore larger mean differences were expected (with smaller deviations) and so $n = 4$ cats were deemed powerful enough to capture all measures. Female cats were randomly assigned to treatment and breeding groups; however, investigators were not blinded to group allocations.

Follicle counts. Seventeen feline ovarian specimens were blinded in terms of treatment and control groups. When possible, the ovarian tissue was bisected to provide two sections of the same ovary on one slide. Average numbers of primordial, primary/secondary, preantral, small antral, large antral, and atretic follicles, as well as corpora lutea, were analyzed using one-way ANOVA with Tukey's post hoc test. Significance is shown when $P < 0.05$.

## Reporting summary

Further information on research design is available in the Nature Portfolio Reporting Summary linked to this article.

## Data availability

A reporting summary and a source data file, containing the principle data presented in this article and its Supplementary Information, are available. The accession codes and corresponding hyperlinks for the AMH sequences referred to in this study are (feline: XP_011286375.2; https://www.ncbi.nlm.nih.gov/protein/XP_011286375.2) and (human: AAH49194.1; https://www.ncbi.nlm.nih.gov/protein/AAH49194.1). Source data are provided with this paper.

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

## Acknowledgements

We thank Christopher Premanandan, DVM, PhD (Ohio State University) for pathology support. Project Grant funding from the National Health and Medical Research Council Australia (1157680, awarded to K.L.W. and C.A.H.) and the Australian Research Council (DP220101179, awarded to

K.L.W and C.A.H.) supported this work. Australian Government Research Training Program scholarships supported S.M. and H.L.

## Author contributions

W.A.S., M.W., M.H., S.B., K.L.W., and C.A.H. conceived and designed the research. W.A.S., L.O., S.M., D.S., H.L., S.G.H, M.W. and A.T. performed the research. W.A.S., L.O., D.S., A.T., M.H., S.B., K.L.W., and C.A.H. analyzed and interpreted the data. W.A.S. and C.A.H. wrote the manuscript. All authors were involved in revising the manuscript.

## Competing interests

Scout Bio provided financial research support and reagents for this project and employees were involved in the conceptualization, oversight and analysis of the in vivo study. L.O., D.S., M.W., A.T., M.H. and S.B. were employees and shareholders of Scout Bio, Inc, at the time the company sponsored this research project. K.L.W. and C.A.H. are inventors of the U.S. patent application number 17/773529 covering the generation of the anti-Müllerian hormone polypeptides used in this manuscript. The authors declare that they have no other competing interests.
