## [Peer Review file · Nature Communications]

Gene therapy with feline anti-Müllerian hormone analogs disrupts folliculogenesis and induces pregnancy loss in female domestic cats

Corresponding Author: Dr William Stocker

Version 0:

Reviewer comments:

Reviewer #1

(Remarks to the Author)

This is an interesting paper that describes the use of engineered feline AMH, expressed from an AAV injected into muscle, as a long-term female contraceptive in cats. The key results are very clear, even in the context of relatively modest numbers of animals tested. The results are also important by virtue of the fact that they highlight the complexity of the effects of AMH on ovarian physiology, a topic that is only just beginning to be explored, particularly in the context of overexpression experiments.

This work derives particular significance as it follows a recent publication, also in Nature Communications, which also showed female contraceptive efficacy of a similar (though not identical) AMH transgene in cats. The bottom-line conclusions of these two studies (as well as that of an earlier mouse study) are similar: female infertility can be brought about by increasing AMH levels. However, the data presented in the three papers have suggested different mechanisms and sites of action. Understanding how AMH works to regulate ovarian function is important from a basic biology perspective, and only a very limited number of species have been explored *in vivo*. The consequences of its up- or down-regulation are also important in multiple clinical contexts, including long-term contraception. This manuscript contributes in important ways to this growing literature.

Here I focus on the two cat studies.

The paper should be published, and it is not plausible to carry out additional experiments (though it would be interesting to know if any blood samples remaining could be used to quantify luteinizing hormone (LH) levels; see below). My primary thoughts revolve around how the current paper can best allow the reader to see how both groups of authors came to their respective conclusions regarding stages regulated by AMH, and the mechanisms that bring about infertility. In addition, it would be useful if the authors could expand a bit more specifically on what future experiments need to be done to provide further insight into how AMH is regulating ovarian physiology and fertility in female cats.

To summarize, the earlier Vansandt et al study used a number of indirect, hormone-based assays to infer the ovarian stages that were altered by AAV-based AMH overexpression over time. To summarize a lot of data, ovarian cycling appeared to not be perturbed. However, they inferred that ovulation probably did not occur in the treated animals. This is primarily based on observing a reduced frequency of prolonged increases in progesterone, which are associated with a luteal phase. However, the fact that some excursions did occur suggest this is not an absolute block.

In mating assays, the Vansandt paper found that treated animals were quite different from controls (though the total numbers are relatively small). While 6 controls were receptive to mating, became pregnant and delivered, only two of the treated females were receptive to mating. Though for one female only two breeding attempts were recorded. These two mated females did not show evidence of pregnancy using ultrasound, and did not deliver any kittens. All of this, combined with the reduced frequency of progesterone excursions, is consistent with AMH action pre-ovulation. That said, most treated females (4) were not receptive and only one underwent many bouts of mating, while a modest number of progesterone excursions (evidence of ovulation?) were observed in the group as a whole, leaving the exact basis of infertility unclear (and Vandsant

acknowledge that their proposed mechanism of action is just their best guess with the data they have). Finally, in the Vansandt study ovaries were not removed and processed for histology and so there is no definitive evidence one way or the other for the occurrence of ovulation (corpus luteum). For similar reasons they were unable to measure the frequency of the various follicle stages directly, or search for other abnormalities associated with increased AMH levels. Thus, an important question becomes whether these observations can be interpreted differently, in terms of mechanism. Alternatively, is there something about the current work (the use of a version of AMH with increased affinity for the receptor, expression level, for example) that could explain the differences.

In the current manuscript, by Stocker et al, the authors first characterized, in vitro and in vivo, an engineered version of feline AMH, designed to be processed efficiently, and to bind more tightly to the AMH receptor. The evidence presented shows AMH is processed efficiently, in vitro and (later in paper) in vivo, and that it has the expected activity in several different assays. These results are clearly presented. The increase in activity of the mutated version of the protein is modestly higher in in vitro assays.

Based on these results the authors introduced AAV that express their protein and followed expression over time. They also measured estrogen and progesterone at regular intervals to gain insight into estrus cycling and likely ovulation events. Breeding trials (which used a more frequent rotation of males than in the earlier study) were performed. Importantly, pregnancies in the treated group were common (as assayed by ultrasound), though none were carried to term. Perhaps most importantly, the authors characterized single ovaries histologically from each cat late in the study. From this analysis it is clear that ovulation occurs, based on the presence of multiple corpus lutea. The frequency of small antral follicles was reduced while those of earlier stages (primordial, primary, preantral) and later stages (large antral) were not. As the authors note, these measurements were taken some time after the end of the breeding study, and thus the timepoints cannot be compared directly. Nonetheless, these histological observations, including the evidence for ovulation, suggest that while AMH reduces the transition from small antral to large, when large antral follicles manage to escape this block (unknown mechanism) they continue to develop through ovulation and corpus luteum formation. Why pregnancies are lost is unclear, but the authors note that related phenotypes have been observed in a mouse model of low level AMH overexpression, which is associated with a failure to downregulate LH during pregnancy, and that LH overexpression can itself induce embryo resorption or miscarriage in mice. Interestingly, Vansandt observed a very significant increase in LH levels in the treated group, though the relationship of this timepoint to other concurrent reproductive events is unknown. The Stocker manuscript did not look at LH levels during pregnancy, leaving its potential significance unclear but easily tested in future mating trials.

Finally, Stocker noted a high frequency of ovarian cysts in the treated group, though the relationship of these cysts to earlier events could not be determined.

A few suggestions.

1. In the abstract the authors express concern about their observation of ovarian cysts in roughly half the treated animals, and their presence being a potential contra-indicator for a long-term contraceptive. However, in the main text (discussion) they note that while the presence of such cysts in guinea pigs can result in several kinds of morbidity, in cats such cysts are typically present as incidental findings in otherwise (apparently) healthy animals. Since the focus here is on cats, not guinea pigs, is it appropriate to comment on the cysts with this negative connotation? If it is, it would be useful to provide a bit more discussion of the point in the discussion. As it stands, the discussion makes it seem that while they are found at some frequency in the general population, they are generally considered benign.

In a related vein, is the presence of a high frequency of pregnancies, which albeit do not come to term, of concern for feral population control. One can easily imagine that owners of cats might find this disconcerting, as well as the fact that their queens are still ovulating, with all its attendant behavioral issues. But does it raise animal welfare or related concerns?

2. In the section referencing Zhou (line 46), the authors might qualify the statement with something like "at least in mice", since they will revisit the topic in their own work.

I would suggest that similar species identifiers be noted at other points in the introduction and elsewhere for various hormones and their effects, since one of the points of this paper is there is significant variability across species.

3. In the section beginning on line 68, where Vansandt is discussed, the results are presented as though the effects on follicular growth phases (such as early antral) are definitively known, based on the hormone assays carried out. Is this the right way to outline the results, given what comes later? Or would it make more sense to more directly lay out a clear outline of what Vansandt measured, why it was a reasonable thing to do for the relevant stages, and how these were used to infer ovarian behavior in the absence of confirmatory histology. The point is to highlight that while Vansandt did not have histology, the inferences they made make sense and are supported by the literature.

4. This is just a question to the authors. Is there any evidence, based on study of related receptors, that altering receptor ligand affinities might lead to differential activation of downstream signaling pathways. In essence, the question is whether there is potentially an interesting mechanistic reason for the differing results, particularly given that AMH and/or its receptors are expressed in diverse tissues.

5. In the first section of the results, where the versions of AMH are being characterized, can the authors present a clear comparison with the Vansandt protein in terms of sequence. Its not quite clear if the two versions used here are identical (other than the single amino acid change meant to alter receptor affinity) or not.

6. In lines 295-298 the authors suggest that AMH may be acting by downregulating FSH receptor expression on antral follicle cells. As noted above, could they indicate that reference 8 is for humans. If this is a clear effect in other species (the review, reference 9), perhaps indicate this more explicitly. The question is whether the reader can reasonably expect the result to generalize to cats from the species that have been characterized, based on universality thus far. A second question has to do with, if this is an important step, why does it take so long to manifest itself? It seems odd that it is showing up so late.

7. In the last part of the discussion the authors note some possible sources of differences in results with Vansandt, and also note that further research is needed to figure it all out. With respect to processing and activity, and related to my earlier question, (and this is just a question for the authors) is there reason to imagine, based on comparisons with other receptor/ligand pairs for this family, that simple quantitative differences in receptor activation may activate distinct pathways, perhaps in different tissues, such as the nervous system versus the ovary. If such behavior has been noted it would be worth mentioning.

Finally, it would be useful if the authors could spend a final paragraph outlining a bit more the limitations of the study and what the next experiments would be that could help to tease out what is going on. They hint at some of this in the discussion of LH, but do not end with anything more specific that would help the reader wrap their head around a specific set of hypotheses that take the observations of both groups and move forward.

Bruce Hay

Reviewer #2

(Remarks to the Author)

In this study the authors aimed to replicate the study of Vansandt et al from 2023, showing the potential of AMH as a contraception in cats. Similar to the previous study the authors used a vector-based approach to produce high AMH levels in cats. While the approach is comparable, importantly more cats were treated. Interestingly, the results of the current study differ from the previous study as in this study treated cats did become pregnant although did not yield a live birth. Independent replications are important, but also to decipher what may explain observed differences. Therefore, additional experiments may be needed.

Comments:

1. In the introduction, the authors nicely highlight differences in effects of AMH between species. It would be useful to add that also loss of AMH results in increased follicular atresia, suggesting that in the presence of both high and low AMH levels counterregulatory mechanisms are activated.
2. Study by the group of Giacobini have shown that AMH also has extra-ovarian functions, i.e., affecting GnRH pulsatility in the hypothalamus and thereby LH secretion, and aromatase expression in the placenta. Given the results presented in this study it would be especially useful to analyze hypothalamic and placental function. This would strengthen the current study as it may provide an explanation for the observed results. For instance, FSH and LH levels were not measured. In mice, increased LH levels result in increased androgen levels. In the placenta, AMH inhibits aromatase expression; consequently, androgens are not "inactivated" to estrogens. Given that particularly pregnancy was affected in the current study, it would be especially useful to assess LH and androgen levels prior to pregnancy. Also, hormone measurements during pregnancy would be useful to gain insight into the underlying mechanism of failed live birth upon supraphysiological levels of AMH.
3. The authors have used two forms of AMH. Similar to the study of Vansandt et al, an optimized cleavage site was introduced, and additionally a mutation was introduced that increases AMHR2 binding and activity. However, based on the results in vivo, there is not a clear difference between both forms. In vivo results nor the in vitro assay using serum from cats showed a difference. How to the authors explain this?
4. It would be useful to show AMH levels in the control cats during the study period. Perhaps as an insert in figure 2c?
5. The authors have analyzed AMH processing in vivo by Western blot. While the same cat was analyzed prior to injection, it would be useful to analyze the same cat over time. For instance, Cat# 987 seems to have lower AMH levels at day 35 than cat #008. This may be a difference between cats, but levels may also differ overtime.
6. An important figure is the analysis of the estrous cycle of the cats. Given that cats can have spontaneous ovulations, these graphs are not easy to interpret. It would be useful if the estrous cycle were assessed in the same cat prior to treatment and after treatment.
7. The authors did not observe a difference in the number of primary and preantral follicles. Only at the small antral stage a significant decrease was observed in the treated animals. Based on the legends, the groups size is n=6, 5 and 6 for the control, RKKR and RKKR/G561 groups respectively. However, based on the graphs these are 10, 6, and 8. Please explain this difference, as it was indicated that the treatment groups, were 6, 6 (-1 for no AMH expression), and 6. Secondly, when carefully analyzing the graph of the large antral follicles, the control group also contain ovaries with no large antral follicles. Can the stage of the estrous cycle influence follicle count and could this impact the obtained results?
8. Given that the authors have isolated the ovaries of the animals it would be useful to assess expression of steroidogenic enzymes (particularly aromatase) in these ovaries by IHC, if proper antibodies are available.

Reviewer #3

(Remarks to the Author)

As an alternative to castration or hysterectomy, AMH has been proposed as a treatment for contraception in domestic animals. Based on previous studies of their group, Stocker and colleagues aimed to generate potent AMH analogues and use them as contraceptives in female cats. Unexpectedly, although supraphysiologic serum AMH was achieved and this provoked a progressive decline in antral follicle numbers and in oestrus cyclicity, ovulation was not completely inhibited; furthermore, elevated AMH was associated with an increased occurrence of non-follicular ovarian cysts. The authors raise their concerns about the use of AMH for fertility control.

The unexpected findings in this feline animal model raise a controversy, challenging the conclusions obtained using other in vitro or in vivo experimental strategies that propose that AMH precludes full folliculogenesis in feline and rodents. The concern raised by the present work is adequately supported by the results and merits attention. Nonetheless, the work needs to go into further details in order to make an outstanding contribution to the knowledge of the physiology and potential therapeutic use of AMH in ovarian folliculogenesis.

Major issues:

1. Experimental design and conclusions:

- a. Was $n=6$ cats per experimental group a sufficient sample size. Indeed, the interindividual variability cannot be appreciated with the present way of reporting the results (see below), and 1 of 6 cats did not respond according to the authors. What proportion would be expected to not respond? Answering this is essential to appraise the efficacy of a treatment, as recommended by the ARRIVE guidelines.
- b. Could the interindividual variability in serum AMH explain the variable impact on folliculogenesis in the treatment groups?
- c. Were the animals in both groups checked for the variability in their oestral cycles and proven fertility previously to treatment administration?
- d. In cats, normally ovulation is induced after copulation. In all groups, there is a significant proportion of cats with spontaneous ovulation. Can the authors explain this phenomenon? Can treatment efficacy vary according to these differences?
- e. In summary, there are several concerns about the variability in both groups (in addition to the previous comments, see e.g. Figures 4 and Suppl 3)

2. Please give the precise (mean/median \pm SD or IQR) serum AMH levels in numbers in the control and experimental groups.

- a. Bars shown in Figure 2 b do not give a precise idea of AMH levels, and it is difficult for the reader to appraise the magnitude change after treatment (7000-fold) referred in the text (line 130). Also please note that the Y axis of Figure 2 b should be improved (ng/ml instead of $\mu\text{g/ml}$) and more ticks in the axis, not just 0.004 and 0.008 $\mu\text{g/ml}$ (which would better be 4 and 8 ng/ml).
- b. Similarly, the Y axes on Figures 2c and 2d would better be in ng/ml, to which the reader is used for physiologic ranges. The eventual need to express AMH as 1000 to 32000 ng/ml conveys better the idea of supraphysiologic levels. Also, I am not sure that using logarithmic scales is helpful. The differences between the individual experimental animals cannot be easily judged. Finally, which criteria were used to exclude the fAMH_RKKR cat that failed from the graph and the analysis? The principle of intent-to-treat should be used.

3. On which basis did the authors decide to analyse the impact of the experimental approach only until 126 days? Could a better contraceptive efficacy be obtained by prolonging the observation period, e.g. if the effect of supraphysiologic AMH was not acute but a chronic one, needing time to be effective? Could the authors also explain the rationale for measuring AMH levels at days 0, 35, 42, 49, 126 and 239?

4. It would be important to show the underlying pathophysiology of the observations:

- a. What explains the disruption of folliculogenesis in these experimental models (fAMH_RKKR and fAMH_RKKR/G561S)? Although the mechanisms could be hypothesised from what is known in the literature, some data supporting this supposition would be desirable.
- b. Similarly, what explains oestradiol synthesis: a decline in aromatase expression, a decrease in granulosa cell numbers, other?

5. Histology:

- a. Quantification of follicular structures: did the authors use adequate morphometric methods to reflect the effects the treatment on the whole ovaries?
- b. In the treated groups, the aspect of the ovaries (Figure 5) looks damaged rather than with an inhibition of folliculogenesis. Can the authors elaborate on this?

Minor issues:

6. The interpretation that the RKKR strategy enhanced processing and the accumulation of mature fAMH \sim 8-fold (line 105 and Fig. 1c) does not seem adequate. Fig 1c reports the results of a Western blot analysis, which is not quantitative. The authors may conclude that the intensity of the signal is 8-fold higher, which does not mean that there is 8-fold higher concentration of mature fAMH. Please revise.

7. Could the authors explain in further detail the luciferase reporter assay?

8. Have the authors assessed receptivity behaviour of female cats in all groups?

9. Please describe in further detail the strategy for mating. How did the authors define the chronology of rotation among the groups: was it based on hormone levels, was it random?

10. The Discussion section is rather lengthy and lacks focus, which undermines the impact of the conclusions. Following the recommendations of the ARRIVE guidelines would certainly help to improve this section.

Reviewer #4

(Remarks to the Author)

I co-reviewed this manuscript with one of the reviewers who provided the listed reports. This is part of the Nature Communications initiative to facilitate training in peer review and to provide appropriate recognition for Early Career Researchers who co-review manuscripts

Version 1:

Reviewer comments:

Reviewer #2

(Remarks to the Author)

The authors have addressed my questions satisfactory. Where necessary they have added additional experiments. Although further studies are needed to unravel the underlying mechanism(s) of AMH action as a potential contraceptive in cats, additional studies at this stage do not seem justified.

However, although addressed adequately in their rebuttal, two answers are not incorporated in the revised manuscript. I suggest correcting these omissions.

1) In follow up of my previous comment 7, it is common to average the number of follicles of two ovaries to reflect the number of animals. Based on their answer, I assume section of two ovaries were counted for some of the cats? The authors are advised to show the average. If the authors have a justified reason to omit this, at least correct the legend to reflect the data points in the graphs.

2) The authors are also advised to add analysis of placental function as part of further research, given that impact of supraphysiological AMH levels is not only impacting the ovary.

Reviewer #3

(Remarks to the Author)

In this revised version, the authors have clarified many of the concerns raised during the assessment of the original version.

Some concerns still remain:

1. The explanation of the sample size (1.a of previous version) does not address the concern. Sample size calculation needs to be performed according to methodological guidance, rather than comparing with other studies. In other words, the inadequacy of a previous study does not justify the lack of sample size calculation in this study. For the reader, it is essential to know whether the sample size was sufficient to support the conclusions of the study with acceptable alpha and beta errors.

2. Regarding the provision of experimental data supporting the mechanism hypothesised on the underlying pathophysiology of the observations ("What explains the disruption of folliculogenesis in these experimental models (fAMH_RKKR and fAMH_RKKR/G561S)?") (4.a of previous version), the authors give more theoretical explanations but do not provide any experimental data.

3. The explanation regarding the decrease in oestradiol production should be included in the manuscript, using seminal references by Viger and di Clemente on the effect of AMH on aromatase (years 1980-1990).

4. The morphometric technique for quantifying ovarian follicles (5.a of previous version) needs to be reported in detail for the reader to be able to critically assess the validity of the conclusions.

Reviewer #4

(Remarks to the Author)

Response to reviewer comments

Reviewer #1 (Remarks to the Author):

This is an interesting paper that describes the use of engineered feline AMH, expressed from an AAV injected into muscle, as a long-term female contraceptive in cats. The key results are very clear, even in the context of relatively modest numbers of animals tested. The results are also important by virtue of the fact that they highlight the complexity of the effects of AMH on ovarian physiology, a topic that is only just beginning to be explored, particularly in the context of overexpression experiments.

This work derives particular significance as it follows a recent publication, also in Nature Communications, which also showed female contraceptive efficacy of a similar (though not identical) AMH transgene in cats. The bottom-line conclusions of these two studies (as well as that of an earlier mouse study) are similar: female infertility can be brought about by increasing AMH levels. However, the data presented in the three papers have suggested different mechanisms and sites of action. Understanding how AMH works to regulate ovarian function is important from a basic biology perspective, and only a very limited number of species have been explored in vivo. The consequences of its up- or down-regulation are also important in multiple clinical contexts, including long-term contraception. This manuscript contributes in important ways to this growing literature.

Here I focus on the two cat studies.

The paper should be published, and it is not plausible to carry out additional experiments (though it would be interesting to know if any blood samples remaining could be used to quantify luteinizing hormone (LH) levels; see below). My primary thoughts revolve around how the current paper can best allow the reader to see how both groups of authors came to their respective conclusions regarding stages regulated by AMH, and the mechanisms that bring about infertility. In addition, it would be useful if the authors could expand a bit more specifically on what future experiments need to be done to provide further insight into how AMH is regulating ovarian physiology and fertility in female cats.

To summarize, the earlier Vansandt et al study used a number of indirect, hormone-based assays to infer the ovarian stages that were altered by AAV-based AMH overexpression over time. To summarize a lot of data, ovarian cycling appeared to not be perturbed. However, they inferred that ovulation probably did not occur in the treated animals. This is primarily based on observing a reduced frequency of prolonged increases in progesterone, which are associated with a luteal phase. However, the fact that some excursions did occur suggest this is not an absolute block.

In mating assays, the Vansandt paper found that treated animals were quite different from controls (though the total numbers are relatively small). While 6 controls were receptive to mating, became pregnant and delivered, only two of the treated females were receptive to mating. Though for one female only two breeding attempts were recorded. These two mated females did not show evidence of pregnancy using ultrasound, and did not deliver any kittens. All of this, combined with the reduced frequency of progesterone excursions, is consistent with AMH action pre-ovulation. That said, most treated females (4) were not receptive and only one underwent many bouts of mating, while a modest number of

progesterone excursions (evidence of ovulation?) were observed in the group as a whole, leaving the exact basis of infertility unclear (and Vandsant acknowledge that their proposed mechanism of action is just their best guess with the data they have). Finally, in the Vansandt study ovaries were not removed and processed for histology and so there is no definitive evidence one way or the other for the occurrence of ovulation (corpus luteum). For similar reasons they were unable to measure the frequency of the various follicle stages directly, or search for other abnormalities associated with increased AMH levels. Thus, an important question becomes whether these observations can be interpreted differently, in terms of mechanism. Alternatively, is there something about the current work (the use of a version of AMH with increased affinity for the receptor, expression level, for example) that could explain the differences.

In the current manuscript, by Stocker et al, the authors first characterized, *in vitro* and *in vivo*, an engineered version of feline AMH, designed to be processed efficiently, and to bind more tightly to the AMH receptor. The evidence presented shows AMH is processed efficiently, *in vitro* and (later in paper) *in vivo*, and that it has the expected activity in several different assays. These results are clearly presented. The increase in activity of the mutated version of the protein is modestly higher in *in vitro* assays.

Based on these results the authors introduced AAV that express their protein and followed expression over time. They also measured estrogen and progesterone at regular intervals to gain insight into estrus cycling and likely ovulation events. Breeding trials (which used a more frequent rotation of males than in the earlier study) were performed. Importantly, pregnancies in the treated group were common (as assayed by ultrasound), though none were carried to term. Perhaps most importantly, the authors characterized single ovaries histologically from each cat late in the study. From this analysis it is clear that ovulation occurs, based on the presence of multiple corpus lutea. The frequency of small antral follicles was reduced while those of earlier stages (primordial, primary, preantral) and later stages (large antral) were not. As the authors note, these measurements were taken some time after the end of the breeding study, and thus the timepoints cannot be compared directly. Nonetheless, these histological observations, including the evidence for ovulation, suggest that while AMH reduces the transition from small antral to large, when large antral follicles manage to escape this block (unknown mechanism) they continue to develop through ovulation and corpus luteum formation. Why pregnancies are lost is unclear, but the authors note that related phenotypes have been observed in a mouse model of low level AMH overexpression, which is associated with a failure to downregulate LH during pregnancy, and that LH overexpression can itself induce embryo resorption or miscarriage in mice. Interestingly, Vansandt observed a very significant increase in LH levels in the treated group, though the relationship of this timepoint to other concurrent reproductive events is unknown. The Stocker manuscript did not look at LH levels during pregnancy, leaving its potential significance unclear but easily tested in future mating trials.

Finally, Stocker noted a high frequency of ovarian cysts in the treated group, though the relationship of these cysts to earlier events could not be determined.

A few suggestions.

1. In the abstract the authors express concern about their observation of ovarian cysts in roughly half the treated animals, and their presence being a potential contra-indicator for a long-term contraceptive. However, in the main text (discussion) they note that while the presence of such cysts in guinea pigs can result in several kinds of morbidity, in cats such cysts are typically present as incidental findings in otherwise (apparently) healthy animals. Since the focus here is on cats, not guinea pigs, is it appropriate to comment on the cysts

with this negative connotation? If it is, it would be useful to provide a bit more discussion of the point in the discussion. As it stands, the discussion makes it seem that while they are found at some frequency in the general population, they are generally considered benign.

We accept the reviewer's point here and have modified the third last and last sentences of the abstract to remove the negative connotation associated with the observed increase in non-follicular ovarian cyst formation in AMH-treated cats.

Third last sentence of abstract (lines 29-32, in manuscript version with mark-up) changed from: "High serum anti-Müllerian hormone caused a progressive decline in antral follicle numbers and was associated with abnormal estrus cyclicity, however, the few surviving large follicles continued to ovulate," to: "High serum anti-Müllerian hormone was associated with abnormal estrus cyclicity, non-follicular ovarian cyst formation and a progressive decline in antral follicle numbers, however, the few surviving large follicles continued to ovulate.

Last sentence of abstract (lines 34-38, in manuscript version with mark-up) changed from: "As elevated anti-Müllerian hormone was also associated with a significant increase in non-follicular ovarian cyst formation, the use of this growth factor for fertility control in companion and free-roaming cats should proceed with caution," to: "Our findings highlight the complexity of the effects of anti-Müllerian hormone on ovarian physiology, but confirm that this growth factor is a candidate for fertility control in free-roaming cats."

In a related vein, is the presence of a high frequency of pregnancies, which albeit do not come to term, of concern for feral population control. One can easily imagine that owners of cats might find this disconcerting, as well as the fact that their queens are still ovulating, with all its attendant behavioral issues. But does it raise animal welfare or related concerns?

To the best of our knowledge, the presence of a high frequency of pregnancies, which do not come to term, does not raise animal welfare or related concerns for use in feral population control. Indeed, conception, followed by pregnancy loss, may actually be beneficial for population control in feral animals.

2. In the section referencing Zhou (line 46), the authors might qualify the statement with something like "at least in mice", since they will revisit the topic in their own work.

As requested, we have modified the sentence (lines 48-50, in manuscript version with mark-up) to read: "However, a recent report by Zhou and colleagues ² indicated that, at least in mice, AMH constantly induces primary follicle atresia in the normal functioning ovary and that preventing the antral follicle pool from becoming too large is this growth factor's primary function."

I would suggest that similar species identifiers be noted at other points in the introduction and elsewhere for various hormones and their effects, since one of the points of this paper is there is significant variability across species.

As requested, we have added the following species identifiers at line 48 “at least in mice”, line 52 “In human granulosa cells and macaque follicles,” line 329 “human” and line 331 “mouse and macaque” (Note: line numbers refer to manuscript version with mark-up shown).

3. In the section beginning on line 68, where Vansandt is discussed, the results are presented as though the effects on follicular growth phases (such as early antral) are definitively known, based on the hormone assays carried out. Is this the right way to outline the results, given what comes later? Or would it make more sense to more directly lay out a clear outline of what Vansandt measured, why it was a reasonable thing to do for the relevant stages, and how these were used to infer ovarian behavior in the absence of confirmatory histology. The point is to highlight that while Vansandt did not have histology, the inferences they made make sense and are supported by the literature.

We agree with this suggestion. As such, we have modified lines 73-85 of the introduction (Note: line numbers refer to manuscript version with mark-up shown).

from:

“Specifically, overexpression of fAMH in cats did not inhibit preantral or early antral follicle growth, as was observed in mice, as levels of estradiol, inhibin A, inhibin B and testosterone did not change, and estrus cyclicity remained unaffected. Rather, based on changes in serum luteinizing hormone and faecal progesterone levels, as well as luteal phase frequency, the authors proposed that high levels of AMH inhibited breeding-induced ovulation, resulting in durable contraception in the cat”

to:

“Specifically, overexpression of fAMH in cats did not appear to inhibit preantral or early antral follicle growth, as was observed in mice, as levels of serum inhibin B (a marker of the growing follicle pool) and faecal estradiol (an antral follicle marker) did not change. The levels of other ovarian hormones, including inhibin A and testosterone, as well as estrus cyclicity, were similarly unaffected by increased circulating levels of fAMH. Rather, based on increased serum luteinizing hormone, decreased faecal progesterone and reduced luteal phase frequency, the authors proposed that high levels of AMH inhibited breeding-induced ovulation, resulting in durable contraception in the cat.”

4. This is just a question to the authors. Is there any evidence, based on study of related receptors, that altering receptor ligand affinities might lead to differential activation of downstream signaling pathways. In essence, the question is whether there is potentially an interesting mechanistic reason for the differing results, particularly given that AMH and/or its receptors are expressed in diverse tissues.

TGF- β superfamily proteins form complexes with type I and type II receptors on the surface of target cells. Once these complexes form, the intracellular kinase domain of the type II receptor phosphorylates and activates the type I receptor. Activated type I receptors then phosphorylate Smad transcription factors (Smad1/5 or Smad2/3), which in turn, form a complex with the co-activator, Smad4. The resulting Smad oligomer translocates to the nucleus and regulates gene transcription.

Over the past two decades, we have generated potent analogues of numerous TGF- β proteins by modifying receptor binding epitopes. These modifications enhance affinity for type I or type II receptors and increase signalling via canonical Smad signalling pathways. We have never observed, for example, AMH analogues activating the alternate Smad2/3 pathway or GDF9 analogues activating the alternate Smad1/5 pathway. Nor have we identified any non-canonical signalling pathways differentially activated by the TGF- β superfamily analogues we have generated. Therefore, we don't anticipate different downstream signalling between AMH variants to be responsible for the differing results between the two studies.

In support, we utilised two fAMH variants in this study: (1) AMH_RKKR, which possesses an enhanced cleavage site, relative to wild-type AMH (but no change to the receptor binding epitopes) and (2) AMH_RKKR/G561S, which possesses both an enhanced cleavage site and increased affinity for the AMH type II receptor, relative to wild-type AMH. The fact that overexpression of both fAMH variants had the same *in vivo* effects (i.e., abnormal estrus cyclicity, a progressive decline in antral follicle numbers, pregnancy, but no live births) indicates that enhancing the affinity of AMH for AMHR2 does not change downstream signalling.

5. In the first section of the results, where the versions of AMH are being characterized, can the authors present a clear comparison with the Vansandt protein in terms of sequence. Its not quite clear if the two versions used here are identical (other than the single amino acid change meant to alter receptor affinity) or not.

The form used by Vansandt *et al* was the unmodified feline AMH protein sequence (depicted in Fig. 1a, with the full amino acid sequence of this protein compared to the human AMH sequence in Supplementary Fig. 1), which we subsequently modified at the cleavage site (A477K/Q478K) to generate the fAMH_RKKR construct. We then added the G561S substitution at the AMHR2 binding site to generate the fAMH_RKKR/G561S construct.

We have included the following sentence at the beginning of the results section (lines 109-111, in manuscript version with mark-up):

“Wild-type fAMH (Fig. 1a and Supplementary Fig. 1), which was overexpressed by Vansandt *et al.* to induce contraception in domestic cats¹⁹, is poorly processed (Fig. 1c) and has low signalling activity (Fig. 1d).”

6. In lines 295-298 the authors suggest that AMH may be acting by downregulating FSH receptor expression on antral follicle cells. As noted above, could they indicate that reference 8 is for humans. If this is a clear effect in other species (the review, reference 9), perhaps indicate this more explicitly. The question is whether the reader can reasonably expect the result to generalize to cats from the species that have been characterized, based on universality thus far. A second question has to do with, if this is an important step, why does it take so long to manifest itself? It seems odd that it is showing up so late.

As requested, we have modified the statement (now lines 329-333, in the manuscript version with mark-up. Note: reference 8 is now 9, and reference 9 is now 10 in the updated manuscript) to read “AMH has been shown to reduce FSH receptor expression in human granulosa cells⁹ and, thereby, diminish FSH-dependent growth and selection of small antral

follicles¹⁰. AMH similarly desensitises growing mouse and macaque follicles to the effects of FSH^{31,32}. These actions may underlie the progressive decline of antral follicles in the feline ovary in the current study.”

In terms of why this effect might take so long to manifest, we don't have a ready answer. However, we can postulate that when AMH is elevated in cats, it impacts survival of the existing cohort of small antral follicles. Reduced survival of this cohort may change ovarian physiology in such a way that fewer follicles progress to the small antral stage in the next wave and so on, until antral follicles are undetectable in AMH-treated animals.

7. In the last part of the discussion the authors note some possible sources of differences in results with Vansandt, and also note that further research is needed to figure it all out. With respect to processing and activity, and related to my earlier question, (and this is just a question for the authors) is there reason to imagine, based on comparisons with other receptor/ligand pairs for this family, that simple quantitative differences in receptor activation may activate distinct pathways, perhaps in different tissues, such as the nervous system versus the ovary. if such behavior has been noted it would be worth mentioning.

As mentioned in response to question 4, we have no evidence from previous studies that enhanced activation of TGF- β receptors will lead to signalling via distinct downstream pathways. Of course, it is possible that increased activation of the canonical Smad1/5 pathway with our more potent AMH analogues, relative to wild-type AMH (used by Vansandt et al.), could underlie the differences between the studies. However, the phenotype observed by Vansandt et al. was, if anything, more “dramatic” (no apparent ovulations, no pregnancies), than the phenotype we observed (ovulation, pregnancy, foetal loss), even though we used more potent versions of AMH.

8. Finally, it would be useful if the authors could spend a final paragraph outlining a bit more the limitations of the study and what the next experiments would be that could help to tease out what is going on. They hint at some of this in the discussion of LH, but do not end with anything more specific that would help the reader wrap their head around a specific set of hypotheses that take the observations of both groups and move forward.

We have shortened the discussion to make it more focused (in response to reviewer 3's suggestion) and have added the following future directions paragraph as the final paragraph.

“Differences in AMH forms used (better processed, more active, in the current study) and the timing/number of breeding trials could partially explain the distinct findings of these studies. However, further research is clearly required to unravel AMH's potential therapeutic or pathologic impact on the feline ovary. Any future study should consider initiating earlier and more frequent breeding trials after AAV:fAMH delivery (e.g., 2-6, 8-12, 14-18 and 20-24 months) to determine how quickly elevated AMH disrupts the maintenance of pregnancy and, potentially, attains contraceptive efficacy. Sufficient serum should also be collected through the pre- and post-treatment periods, including during the breeding trials, to determine how hypothalamic-pituitary-ovarian function is disrupted and, thereby, contributes to the observed pregnancy failure. Although difficult, given the cost of feline studies, performing histology, immunohistochemistry and RNA sequencing on ovaries at different timepoints after AMH

overexpression (e.g., 4, 8 and 12 months) would provide invaluable information on the progressive decline in ovarian function and the formation of ovarian cysts”.

Bruce Hay

Reviewer #2 (Remarks to the Author):

In this study the authors aimed to replicate the study of Vansandt et al from 2023, showing the potential of AMH as a contraception in cats. Similar to the previous study the authors used a vector-based approach to produce high AMH levels in cats. While the approach is comparable, importantly more cats were treated. Interestingly, the results of the current study differ from the previous study as in this study treated cats did become pregnant although did not yield a live birth.

Independent replications are important, but also to decipher what may explain observed differences. Therefore, additional experiments may be needed.

Comments:

1. *In the introduction, the authors nicely highlight differences in effects of AMH between species. It would be useful to add that also loss of AMH results in increased follicular atresia, suggesting that in the presence of both high and low AMH levels counterregulatory mechanisms are activated.*

We have added the following sentence to the Introduction (lines 50-52, in the manuscript version with mark-up):

“Interestingly, mice lacking AMH also display increased follicular atresia, highlighting the critical homeostatic role this hormone plays within the ovary ⁸.”

2. *Study by the group of Giacobini have shown that AMH also has extra-ovarian functions, i.e., affecting GnRH pulsatility in the hypothalamus and thereby LH secretion, and aromatase expression in the placenta. Given the results presented in this study it would be especially useful to analyze hypothalamic and placental function. This would strengthen the current study as it may provide an explanation for the observed results. For instance, FSH and LH levels were not measured. In mice, increased LH levels result in increased androgen levels. In the placenta, AMH inhibits aromatase expression; consequently, androgens are not “inactivated” to estrogens. Given that particularly pregnancy was affected in the current study, it would be especially useful to assess LH and androgen levels prior to pregnancy. Also, hormone measurements during pregnancy would be useful to gain insight into the underlying mechanism of failed live birth upon supraphysiological levels of AMH.*

In response to this suggestion, we measured LH levels in control and AMH overexpressing cats in the limited serum samples we had remaining:

- In serum collected from control ($n=5$ individual cats) and AMH overexpressing ($n=17$ samples, collected from 9 individual cats at different time points) cats during estrus phases between days 0-90 of the study, LH levels were very similar (10.5 ± 6.3 ng/ml vs. 10 ± 4.3 ng/ml, $P = 0.86$) (Supplementary Fig. 5a).
- Interestingly, in serum samples collected at day 126 from AMH overexpressing cats (no control serum was available at this timepoint), LH levels were significantly higher in cats with progressive cycle abnormalities ($n=5$), compared to those with normal cycles ($n=3$) (21.5 ± 3.1 ng/ml vs. 7.7 ± 4.7 ng/ml, $P < 0.01$). This finding suggests that transgenic AMH-induced declines in antral follicle numbers and estrus cyclicity were associated with reduced negative feedback by ovarian steroids to the hypothalamus and anterior pituitary (Supplementary Fig. 5b).

We have generated a new Supplementary Fig. 5 to display this data and describe the results in lines 187-192 and 201-206 of the manuscript (Note: line numbers refer to the manuscript version with mark-up).

Clearly, it would have been beneficial for the study if we had collected enough serum throughout the pre- and post-treatment periods (including during the breeding study) to measure multiple ovarian and pituitary hormones. Unfortunately, we did not anticipate that cats overexpressing our AMH analogues would become pregnant and, therefore, did not prioritise hormone measurements during the second half of the trial. In addition, because pregnancies were lost mid-gestation, we failed to harvest any placental tissue for further analyses.

As cat studies are very expensive (US\$300,000 for the current trial), it is not feasible to repeat the study in order to fill-in these missing pieces of data.

However, the above suggestions are all excellent and those that are feasible should definitely be incorporated into the next feline AMH trial. To this end, we have included an analysis of hypothalamic and pituitary function in a new limitations/future directions paragraph at the end of the discussion (requested by reviewer 1).

3. The authors have used two forms of AMH. Similar to the study of Vansandt et al, an optimized cleavage site was introduced, and additionally a mutation was introduced that increases AMHR2 binding and activity. However, based on the results in vivo, there is not a clear difference between both forms. In vivo results nor the in vitro assay using serum from cats showed a difference. How to the authors explain this?

Vansandt *et al* actually used two doses of unmodified feline AMH in their study (protein sequence depicted in Fig. 1a, with the full amino acid sequence compared to the human AMH sequence in Supplementary Fig. 1). Our fAMH variants incorporated an enhanced cleavage site (fAMH_RKKR) or an enhanced cleavage site with an additional mutation that increases affinity for AMHR2 (fAMH_RKKR/G561S).

In vitro, fAMH_RKKR/G561S is ~4-fold more potent than fAMH_RKKR. We saw the same potency difference when we examined the human versions of these AMH constructs (Stocker et al. FASEB J 2024; 38:e23377). Despite this difference in potency, both fAMH variants produce the same maximal response in vitro.

When delivered *in vivo*, via a single dose of viral vector, the amounts of circulating fAMH_RKKR and fAMH_RKKR/G561S were extremely high (1-20 µg/ml). At these concentrations, subtle potency differences are indiscernible (i.e., concentrations of AMH were well above those needed to generate a maximal response in the ovary or other tissues).

In support of the above, when serum from cats expressing AMH_RKKR or AMH_RKKR/G561S was diluted 1:100, it still displayed identical *in vitro* activity. Further serum dilutions would have confirmed potency differences between the two AMH variants.

Because differences in activity of the AMH variants could not be discerned in serum, we did not make any distinctions between the *in vivo* actions of AMH_RKKR and AMH_RKKR/G561S.

4. It would be useful to show AMH levels in the control cats during the study period. Perhaps as an insert in figure 2c?

AMH levels were measured in control cats at 9 timepoints throughout the study. We have included this data as Fig. 2c. AMH levels did not increase above 6 ng/ml in control cats at any time during the study.

5. The authors have analyzed AMH processing in vivo by Western blot. While the same cat was analyzed prior to injection, it would be useful to analyze the same cat over time. For instance, Cat# 987 seems to have lower AMH levels at day 35 than cat #008. This may be a difference between cats, but levels may also differ overtime.

We have now analysed the *in vivo* secretion and processing of transgenic AMH in 4 cats at 3- to-5 different timepoints between days 0 and 239 of the trial. We have replaced the original Fig. 3a, showing 2 timepoints for each of 4 AMH overexpressing cats, with a new Western blot, showing the expression and processing of transgenic AMH in cat #985 at days 0, 42, 126 and 239. We also analysed day 0 and day 239 serum from control cat #134, which was transduced with empty AAV vector, on the same Western blot. Serum from cat #985 was used in the activity assay in Fig. 3b, so the two panels in this figure are now better aligned. Additional Western blots for AMH-overexpressing cats (#987, #984 and #008) and control cats (#981, #988 and #990) have been incorporated into a new Supplementary Figure 2.

Important note regarding banding patterns in Western blots:

The original Western blot was performed in the US and essentially showed a single mature AMH band in serum from transgenic mice. We repeated the Western blots at Monash University in Australia, using the same primary antibody but a different anti-mouse HRP-conjugated secondary antibody. As with the original Western blot, the new Westerns show that mature AMH (~13 kDa), but not unprocessed AMH precursor (expected size ~75 kDa), was present in serum collected at different timepoints from cats transduced with AAV:fAMH analogues. The intensity of the mature AMH bands closely matches the ELISA data (Fig. 2c-e). No mature AMH was detected in serum collected from cats prior to transduction (day 0), or in control cats transduced with an empty AAV vector.

However, in the Westerns performed in Australia, we observed additional bands between 50-60 kDa, as well as some fainter bands at ~30, ~80 and ~110 kDa. These bands were deemed

non-specific, as they were present in serum collected from cats prior to transduction with AAV:fAMH and in serum from control cats transduced with empty AAV vector. We have indicated the specific and non-specific bands on each Western blot (Fig. 3a and Supplementary Figure 2).

6. An important figure is the analysis of the estrous cycle of the cats. Given that cats can have spontaneous ovulations, these graphs are not easy to interpret. It would be useful if the estrous cycle were assessed in the same cat prior to treatment and after treatment.

We apologise for not being clearer in our description of these graphs. The line on the graph at day 14 represents the timing of AAV vector delivery. Thus, days 0-14 represent the estrous cycle prior to treatment, while days 15-126 represent the estrous cycle after treatment, in each cat. As can be seen in the examples below, cat #984 showed normal estrous cyclicity before and after AAV: AMH delivery, while cat #985 had spontaneously ovulated prior to AAV:AMH delivery, and again around day 48, as evidenced by the sustained increase in serum progesterone measured between days 48-84.

We have altered the sentence (lines 175-177, in manuscript version with mark-up) to read:

“To assess the effect of high levels of active AMH analogs on the feline ovarian cycle, serum progesterone and estradiol levels were measured before (from day 0 to day 14) and after (from day 15-126) AAV delivery of AMH analogues or empty vector.”

7. The authors did not observe a difference in the number of primary and preantral follicles. Only at the small antral stage a significant decrease was observed in the treated animals. Based on the legends, the groups size is n=6, 5 and 6 for the control, RKKR and

RKKR/G561 groups respectively. However, based on the graphs these are 10, 6, and 8. Please explain this difference, as it was indicated that the treatment groups, were 6, 6 (-1 for no AMH expression), and 6.

When possible, the ovarian tissue was bisected to provide two sections of the same ovary on one slide. Where two sections were obtained, follicle numbers for both sections were included in the data set and graph. As such, the number of data points shown on the graph is greater than the number of animals for each group used in the study. In some cases, only one section was able to be obtained, therefore, the number of data points for each group also differs.

Secondly, when carefully analyzing the graph of the large antral follicles, the control group also contain ovaries with no large antral follicles. Can the stage of the estrous cycle influence follicle count and could this impact the obtained results?

The stage of the estrous cycle is likely to affect the number of large antral follicles present in the control ovaries, and likely explains why some of the control cats had no large antral follicles present. As the treated cats however also had no small antral follicles present, that would have had the potential to grow into large antral follicles at the relevant stage of the estrous cycle, this loss of small antral follicles likely explains why no large antral follicles were then present in these cats.

8. Given that the authors have isolated the ovaries of the animals it would be useful to assess expression of steroidogenic enzymes (particularly aromatase) in these ovaries by IHC, if proper antibodies are available.

We have a follow-up study planned to consider the effects of AMH overexpression on gene/protein expression in the feline ovary. The immunohistochemistry component of this study will require significant optimisation, which is outside the scope of the current manuscript. However, we have included this suggestion in the new limitations/future directions paragraph at the end of the discussion.

Reviewer #3 (Remarks to the Author):

As an alternative to castration or hysterectomy, AMH has been proposed as a treatment for contraception in domestic animals. Based on previous studies of their group, Stocker and colleagues aimed to generate potent AMH analogues and use them as contraceptives in female cats. Unexpectedly, although supraphysiologic serum AMH was achieved and this provoked a progressive decline in antral follicle numbers and in oestrus cyclicity, ovulation was not completely inhibited; furthermore, elevated AMH was associated with an increased occurrence of non-follicular ovarian cysts. The authors raise their concerns about the use of AMH for fertility control.

The unexpected findings in this feline animal model raise a controversy, challenging the conclusions obtained using other in vitro or in vivo experimental strategies that propose that AMH precludes full folliculogenesis in feline and rodents. The concern raised by the present work is adequately supported by the results and merits attention. Nonetheless, the work needs to go into further details in order to make an outstanding contribution to the knowledge of the physiology and potential therapeutic use of AMH in ovarian folliculogenesis.

Major issues:

1. Experimental design and conclusions:

a. Was n=6 cats per experimental group a sufficient sample size. Indeed, the interindividual variability cannot be appreciated with the present way of reporting the results (see below), and 1 of 6 cats did not respond according to the authors. What proportion would be expected to not respond? Answering this is essential to appraise the efficacy of a treatment, as recommended by the ARRIVE guidelines.

The study by Vansandt *et al.* (Nature Communications, 2023, 14:3140) used n=3 female cats per group (control, low AMH and high AMH). During each of two 4-month long mating trials, the 3 control females conceived and gave birth to 2-4 healthy kittens. In contrast, no females expressing low levels of AMH (n=3) or high levels of AMH (n=3) conceived/gave birth during either mating trial. Based on these findings, the authors concluded that ectopic expression of AMH resulted in safe and durable contraception in female domestic cats.

As we were delivering more potent AMH analogues than those used in the study by Vansandt *et al.*, we expected to similarly induce complete infertility in female cats. Complete infertility would result in no pregnancies or live births (the primary outcomes measured) in AMH treated female cats, while controls were expected to breed normally. As such, n=6 females per group was deemed sufficient to measure differences in the primary outcomes of the study. NB: this is double the number of cats per group (n=3) used in the study by Vansandt *et al.*

Response rate in this study was determined by a substantial increase in serum levels of AMH following intramuscular delivery of AAV vector. Normally, this is a very robust technique for overexpressing proteins *in vivo* and 100% of cats were expected to respond. For the cat that did not respond following delivery of the AAV-fAMH_RKKR vector, we assume there was an issue during injection, which precluded the vector transducing muscle cells.

b. Could the interindividual variability in serum AMH explain the variable impact on folliculogenesis in the treatment groups?

At the end of the study, treated cats had serum AMH levels between 1 µg/ml (#983) and 14 µg/ml (#008), which were approximately 500-7000 times control levels. While all AMH overexpressing cats had no antral follicles present in their ovaries at the end of the study, no correlation could be made between the concentration of serum AMH in an individual cat and changes in estrous cyclicity (between days 0-126) or ovarian cyst formation. We concluded from these results that circulating AMH levels in all treated cats were above the maximum required to progressively suppress antral follicle survival, but that the timing of these effects (i.e., before or after day 126) and the development of ovarian cysts were variable.

c. Were the animals in both groups checked for the variability in their oestral cycles and proven fertility previously to treatment administration?

In the graphs shown in Fig. 4 and Supp. Fig. 4, days 0-14 represent changes in ovarian steroids prior to treatment in each cat. Nine of the seventeen cats in the study displayed normal estrus cyclicity prior to treatment, as evidenced by a 2- to 3-fold increase in estradiol levels, together with low and relatively stable levels of progesterone. The remainder of the cats experienced a spontaneous ovulation during this pretreatment period, with progesterone levels elevated ~10-fold.

Female cats were of reproductive age, but their fertility had not been confirmed prior to the study. However, fertility was confirmed during the study in 14/17 cats with detected pregnancies. The 3 cats that did not conceive displayed normal cycles or spontaneous ovulatory cycles during the hormone measurement phase of the study and they each had corpora lutea present in their ovaries at endpoint.

d. In cats, normally ovulation is induced after copulation. In all groups, there is a significant proportion of cats with spontaneous ovulation. Can the authors explain this phenomenon? Can treatment efficacy vary according to these differences?

Whilst ovulation in cats is typically induced after copulation, spontaneous ovulations are commonly observed in group-housed cats, as described by Binder and colleagues (*Anim Reprod Sci* 209, 106167, 2019). In control cats, whether they displayed normal estrous cycles (n=3) or repeated spontaneous ovulations (n=3) during days 0-126, did not affect their subsequent ability to conceive or give birth to healthy litters during the breeding trial. Similarly, in AMH overexpressing groups, 4 cats with normal estrous cycles and 4 cats with repeated spontaneous ovulations between days 0-126 conceived during the breeding trial, although none could carry their pregnancies to term. Thus, the data does not support an effect of spontaneous ovulation on treatment efficacy.

e. In summary, there are several concerns about the variability in both groups (in addition to the previous comments, see e.g. Figures 4 and Suppl 3)

In the answers above, we have addressed the reviewer's concerns about variability in serum AMH levels, estrous cyclicity/fertility prior to treatment and spontaneous ovulation rates, as well as whether each experimental group had sufficient numbers. Importantly, variability in these parameters have enabled us to determine that:

1. AMH overexpression will disrupt the maintenance of pregnancy in cats cycling normally or undergoing spontaneous ovulations prior to breeding.
2. AMH levels ~500-fold normal are sufficient to progressively disrupt folliculogenesis in cats.

2. Please give the precise (mean/median \pm SD or IQR) serum AMH levels in numbers in the control and experimental groups.

In Fig. 2 c-e, we have added the mean \pm SD for AMH serum levels in control and treated cats between days 35 and 281.

a. Bars shown in Figure 2 b do not give a precise idea of AMH levels, and it is difficult for the reader to appraise the magnitude change after treatment (7000-fold) referred in the text (line 130). Also please note that the Y axis of Figure 2 b should be improved (ng/ml instead of $\mu\text{g/ml}$) and more ticks in the axis, not just 0.004 and 0.008 $\mu\text{g/ml}$ (which would better be 4 and 8 ng/ml).

We have changed the y-axis label to ng/ml and added additional ticks/numbers.

b. Similarly, the Y axes on Figures 2c and 2d would better be in ng/ml, to which the reader is used for physiologic ranges. The eventual need to express AMH as 1000 to 32000 ng/ml conveys better the idea of supraphysiologic levels. Also, I am not sure that using logarithmic scales is helpful. The differences between the individual experimental animals cannot be easily judged. Finally, which criteria were used to exclude the fAMH_RKKR cat that failed from the graph and the analysis? The principle of intent-to-treat should be used.

We have changed the y-axis label to ng/ml on Fig. 2c-e. We agree with the reviewer that this better reflects the supraphysiological levels of AMH present in treated cats. As stated above, we have added mean \pm SD for AMH serum levels in control and treated cats to each graph, which makes it easier to judge the differences between individual treated animals (and controls).

The aim of the study was to assess the contraceptive efficacy of AMH analogues delivered via AAV vectors to female domestic cats. One cat that was injected intramuscularly with the AAV: fAMH_RKKR vector failed to overexpress AMH (likely due to injection error). As AMH was not elevated in this cat during the first 20 days after treatment, it was excluded from subsequent analyses assessing the effects of supraphysiological levels of AMH on ovarian steroid production, pregnancy and live births. This criterium was established a priori.

3. On which basis did the authors decide to analyse the impact of the experimental approach only until 126 days? Could a better contraceptive efficacy be obtained by prolonging the observation period, e.g. if the effect of supraphysiologic AMH was not acute but a chronic one, needing time to be effective? Could the authors also explain the rationale for measuring AMH levels at days 0, 35, 42, 49, 126 and 239?

Kano *et al.* demonstrated that AMH overexpression in mice led to a complete block in folliculogenesis beyond the primary stage by 40 days (Kano M, *et al.* AMH/MIS as a contraceptive that protects the ovarian reserve during chemotherapy. *Proc Natl Acad Sci U S A* 114, E1688-E1697 (2017)). As the time taken from activation of a primordial follicle to ovulation typically lengthens in species larger than mice, we determined that an initial study length of 126 days (18 weeks) should be sufficient to observe a similar block in folliculogenesis following AMH overexpression in cats. Pleasingly, we did observe late abnormalities in the levels of reproductive hormones in approximately half of the AMH treated cats. A longer analysis period would likely have uncovered a progressive decline in ovarian cycling in additional animals, however, the costs (>\$1500/day) associated with blood collection every two days and clinical chemistry to measure steroid levels were prohibitive.

However, we did wait an additional 10 weeks before initiating the breeding trial, in order to maximise the contraceptive efficacy of transgenic AMH.

Serum AMH levels were measured by ELISA at day 0, day 14 (the day of AAV-delivery) and then weekly (days 21, 28, 35 etc.) until day 98 and fortnightly, thereafter (see Fig. 2d,e). Serum samples assessed by Western blotting to determine AMH processing efficiency *in vivo* were selected based on the time point at which circulating AMH levels were at their highest, as determined by ELISA. The rationale for selecting samples containing the highest AMH concentrations was that processing efficiency could be expected to be at its lowest when the quantity of AMH being synthesised is at its maximum. Processing efficiency could therefore be assumed to be greater when the quantity of AMH being synthesised was lower.

Note: to address a suggestion by reviewer 2 (point 5), we have now analysed by Western blotting transgenic AMH in 4 cats at 3-to-5 different timepoints between days 0 and 239 of the trial, with the timepoints selected based on the availability of remaining serum. We have replaced the original Fig. 3a, showing 2 timepoints for each of 4 AMH overexpressing cats, with a new Western blot, assessing transgenic AMH in cat #985 at days 0, 42, 126 and 239. We also analysed day 0 and day 239 serum from control cat #134, which was transduced with empty AAV vector, on the same Western blot. Additional Western blots for AMH-overexpressing cats (#987, #984 and #008) and control cats (#981, #988 and #990) with a greater number of timepoints than shown in the original Fig. 3a have been incorporated into a new Supplementary Figure 2.

4. It would be important to show the underlying pathophysiology of the observations:

a. What explains the disruption of folliculogenesis in these experimental models (fAMH_RKKR and fAMH_RKKR/G561S)? Although the mechanisms could be hypothesised from what is known in the literature, some data supporting this supposition would be desirable.

Histology, performed 8 months after the final hormone measurements, supports a concept of progressive ovarian dysfunction in the presence of high serum AMH, as antral follicles were undetected in all treated cats by this time (even those that were cycling normally at the end of hormone measurements). AMH has been shown to reduce FSH receptor expression in human granulosa cells and, thereby, diminish FSH-dependent growth and selection of small antral follicles. AMH similarly desensitises growing mouse and macaque follicles to the effects of FSH. These actions may underlie the progressive decline of antral follicles in the feline ovary in the current study. An anticipated consequence of this decline would be reduced negative feedback by ovarian steroids to the hypothalamus and anterior pituitary, and a subsequent increase in gonadotropin (LH and FSH) release. Indeed, in serum samples collected at day 126 from AMH overexpressing cats, LH levels were significantly higher in cats with progressive cycle abnormalities ($n=5$), compared to those with normal cycles ($n=3$) (21.5 ± 3.1 ng/ml vs. 7.7 ± 4.7 ng/ml, $P < 0.01$). (Supplementary Fig. 5b).

b. Similarly, what explains oestradiol synthesis: a decline in aromatase expression, a decrease in granulosa cell numbers, other?

As antral follicle numbers had declined precipitously in AMH overexpressing cats when ovaries were harvested at the end of the study, the mid-study decline in estradiol levels in some animals is most likely due to a reduction in follicular granulosa cells. However, as AMH is known to down regulate aromatase in a number of experimental models (Pellatt *et al.* Anti-Mullerian hormone reduces follicle sensitivity to follicle-stimulating hormone in human granulosa cells. *Fertil Steril* **96**, 1246-1251 e1241 (2011).; Xu *et al.* Anti-Mullerian hormone promotes pre-antral follicle growth, but inhibits antral follicle maturation and dominant follicle selection in primates. *Hum Reprod* **31**, 1522-1530 (2016).), this could also be a contributing factor.

5. Histology:

a. Quantification of follicular structures: did the authors use adequate morphometric methods to reflect the effects the treatment on the whole ovaries?

Quantification of feline ovarian follicular structures was performed using the criteria described by Bristol-Gould and Woodruff (*Theriogenology* 66, 5-13 (2006)).

b. In the treated groups, the aspect of the ovaries (Figure 5) looks damaged rather than with an inhibition of folliculogenesis. Can the authors elaborate on this?

Histology was performed by the Comparative Theriogenology and Reproductive Pathology Service at The Ohio State University. Slides were analysed by Professor Christopher Premanandan (Diplomate, American College of Veterinary Pathologists). Professor Premanandan indicated that the histology of the ovaries was consistent with reduced follicle numbers and/or ovarian cysts compressing the ovarian parenchyma.

Minor issues:

6. The interpretation that the RKKR strategy enhanced processing and the accumulation of mature fAMH ~8-fold (line 105 and Fig. 1c) does not seem adequate. Fig 1c reports the results of a Western blot analysis, which is not quantitative. The authors may conclude that the intensity of the signal is 8-fold higher, which does not mean that there is 8-fold higher concentration of mature fAMH. Please revise.

We have deleted ~8-fold from this sentence.

7. Could the authors explain in further detail the luciferase reporter assay?

HEK293T cells were plated at 1.5×10^4 cells/well in 96-well plates in DMEM supplemented with 10% FCS and incubated at 37°C in 5% CO₂. After overnight incubation, Lipofectamine 2000 (Life Technologies) was used according to the manufacturer's instructions to transfect cells with 100 ng/well of plasmid DNA, consisting of: 4xBRE-Luc (98.9 ng), AMHR2 (0.8 ng) and ALK2 (0.3 ng), diluted in OPTI-MEM. Approximately 24 hours after transfection, cells were treated with AMH variants diluted in serum-free medium (DMEM high glucose supplemented with 1 mM sodium pyruvate (Life Technologies) and 0.01% BSA (Sigma-

Aldrich, St. Louis, MO, United States)) and incubated overnight (~18 hours) at 37°C in 5% CO₂. The medium was then removed and cells lysed in solubilisation buffer (26 mM glycylglycine (pH 7.8), 16 mM MgSO₄, 4 mM EGTA, 900 µM dithiothreitol, 1% Triton X-100). The lysate was transferred to a white 96-well plate (Corning® Costar®, Corning, NY, United States) and luminescence measured immediately after the addition of the substrate luciferin (Promega, Madison, WI, United States), using a CLARIOstar microplate reader (BMG Labtech, Ortenberg, Germany).

8. *Have the authors assessed receptivity behaviour of female cats in all groups?*

For the first 126 days of the study, we assessed behavioural traits of the female cats (vocalisation, tail deflection, treading, rubbing, rolling/Lordosis) to see if we could determine if they were in heat. We assigned scores for each of the behaviours and had a cumulative score that we hoped would be predictive of heat behaviour, however, we did not see anything compelling in the data. Cats were not continually monitored during the breeding stage, but any observations of mounting were recorded. In all, 10 mounting events were observed (5 in control cats, 3 in cat #129 that failed to elevate AMH levels following AAV delivery, 1 in cat #991 from the AAV-fAMH_RKKR group and 1 in cat #984 from the AAV-fAMH_RKKR/G561S group). While this data was suggestive of a reduction in breeding interactions in cats overexpressing AMH (a finding observed by Vansandt *et al.*), constant video monitoring would be required to confirm this observation.

9. *Please describe in further detail the strategy for mating. How did the authors define the chronology of rotation among the groups: was it based on hormone levels, was it random?*

For the breeding portion of the study, 2 female cats from each of the groups were randomly assigned to one of three breeding groups. One of 5 intact males was added to each breeding group on Day 197. The male cats were rotated at least every two weeks so that each of the breeding groups was exposed to each male cat at least once during the 12-week mating trial.

10. *The Discussion section is rather lengthy and lacks focus, which undermines the impact of the conclusions. Following the recommendations of the ARRIVE guidelines would certainly help to improve this section.*

We reduced the length of the original discussion by 2/3 of a page and used the ARRIVE guidelines to better focus this section (e.g., better interpreting the results of the study in the context of the study objectives and further discussing the results in the context of the current theory). Furthermore, we have included a final paragraph (shown below) that describes potential future experiments to progress AMH's therapeutic translation in female cats. This future research paragraph also aligns with the ARRIVE guidelines.

“Differences in AMH forms used (better processed, more active, in the current study) and the timing/number of breeding trials could partially explain the distinct findings of these studies. However, further research is clearly required to unravel AMH's potential therapeutic or pathologic impact on the feline ovary. Any future study should consider initiating earlier and more frequent breeding trials after AAV:fAMH delivery (e.g., 2-6, 8-12, 14-18 and 20-24

months) to determine how quickly elevated AMH disrupts the maintenance of pregnancy and, potentially, attains contraceptive efficacy. Sufficient serum should also be collected through the pre- and post-treatment periods, including during the breeding trials, to determine how hypothalamic-pituitary-ovarian function is disrupted and, thereby, contributes to the observed pregnancy failure. Although difficult, given the cost of feline studies, performing histology, immunohistochemistry and RNA sequencing on ovaries at different timepoints after AMH overexpression (e.g., 4, 8 and 12 months) would provide invaluable information on the progressive decline in ovarian function and the formation of ovarian cysts.”

Reviewer #4 (Remarks to the Author):

I co-reviewed this manuscript with one of the reviewers who provided the listed reports. This is part of the Nature Communications initiative to facilitate training in peer review and to provide appropriate recognition for Early Career Researchers who co-review manuscripts

NCOMMS-24-33876A, “Gene therapy with feline anti-Müllerian hormone analogs disrupts folliculogenesis and induces pregnancy loss in female domestic cats”: Response to reviewer comments

Reviewer #2 (Remarks to the Author):

The authors have addressed my questions satisfactory. Where necessary they have added additional experiments. Although further studies are needed to unravel the underlying mechanism(s) of AMH action as a potential contraceptive in cats, additional studies at this stage do not seem justified.

However, although addressed adequately in their rebuttal, two answers are not incorporated in the revised manuscript. I suggest correcting these omissions.

1) In follow up of my previous comment 7, it is common to average the number of follicles of two ovaries to reflect the number of animals. Based on their answer, I assume section of two ovaries were counted for some of the cats? The authors are advised to show the average. If the authors have a justified reason to omit this, at least correct the legend to reflect the data points in the graphs.

As requested, for animals where two ovarian sections were able to be obtained, we have graphed the mean numbers of each follicle type so that each data point now represents the follicle numbers for an individual animal. As such, the n numbers between the graphs and figure legend now correspond ($n=6, 5$ and 6 : for the control, fAMH_RKKR and fAMH_RKKR/G561S groups, respectively).

2) The authors are also advised to add analysis of placental function as part of further research, given that impact of supraphysiological AMH levels is not only impacting the ovary.

As requested, we have added the following statement to the end of the discussion:

“In pregnant mice, disruption of hypothalamic-pituitary function due to, and together with, elevated AMH alters placental steroid metabolism, with smaller litter sizes observed in these mice due to an increased number of aborted embryos⁴³. Future cat studies should therefore assess placental function, to determine whether this is also a contributing factor to the pregnancy loss we observed in cats following AMH overexpression.”

Reviewer #3 (Remarks to the Author):

In this revised version, the authors have clarified many of the concerns raised during the assessment of the original version.

Some concerns still remain:

1. The explanation of the sample size (1.a of previous version) does not address the concern. Sample size calculation needs to be performed according to methodological guidance, rather than comparing with other studies. In other words, the inadequacy of a previous study does not justify the lack of sample size calculation in this study. For the reader, it is essential to know whether the sample size was sufficient to support the conclusions of the study with acceptable alpha and beta errors.

As requested, we have performed a sample size calculation, and we have added the following statement into the relevant portion of the Methods section (under the section subtitled Statistics):

“Prior studies examining vectored AMH in mice ³ showed complete infertility, and revealed large differences in the number of growing primary, secondary and antral follicles, as well as corpora lutea within the ovaries. Follow-up studies in cats ²¹ led by the same team indicated that vectored AMH delivery in cats also resulted in complete infertility. As such, it was predicted that mean follicle counts and fertility parameters would be vastly different between experimental groups, consequently requiring less animals to reach statistical significance. Based on these studies, we performed power calculations assuming a minimum mean difference between follicle counts of $\geq 50\%$ (with standard deviations of $\geq 25\%$), with a probability of $P < 0.05$ (alpha level 5%), and power of 80% (beta level 20%). Using these values, we estimated that a minimum of $n=4$ cats would be sufficient to reach statistical significance in follicle counts between groups. For fertility measures, we predicted that the AMH vectored contraceptive would induce complete infertility, therefore larger mean differences were expected (with smaller deviations) and so $n=4$ cats was deemed powerful enough to capture all measures.”

2. Regarding the provision of experimental data supporting the mechanism hypothesised on the underlying pathophysiology of the observations (“What explains the disruption of folliculogenesis in these experimental models (fAMH_RKKR and fAMH_RKKR/G561S)?”) (4.a of previous version), the authors give more theoretical explanations but do not provide any experimental data.

We are unable to provide additional experimental data to address this question conclusively in the current study.

3. The explanation regarding the decrease in oestradiol production should be included in the manuscript, using seminal references by Viger and di Clemente on the effect of AMH on aromatase (years 1980-1990).

As requested, we have modified the explanation (lines 48-50 of the revised manuscript) so that the foundational papers (full reference below) are now cited, with reference to AMH causing a decrease in oestradiol production:

9. Vigier B, *et al.* Anti-Mullerian hormone produces endocrine sex reversal of fetal ovaries. *Proc Natl Acad Sci USA* **86**, 3684-3688 (1989).

10. Di Clemente N, *et al.* A quantitative and interspecific test for biological activity of anti-Müllerian hormone: the fetal ovary aromatase assay. *Development* **114**, 721–727 (1992).

4. The morphometric technique for quantifying ovarian follicles (5.a of previous version) needs to be reported in detail for the reader to be able to critically assess the validity of the conclusions.

We have expanded our description of the criteria used within the Methods section to read as follows (the additional detail is underlined below):

The following criteria were utilized to quantify follicular structures ^{49, 50}: (1) Primordial follicles - an oocyte 20 to 30 μ m in diameter, surrounded by a single layer of flattened follicular cells; (2) Primary/Secondary follicles - an oocyte 30 to 75 μ m in diameter with an obvious zona pellucida, surrounded by a single layer or multiple layers of cuboidal cells, and a total follicle diameter of 100 to 400 μ m, depending on the number of granulosa cell layers, but with no evidence of space formation between the granulosa cells. The outermost layer of granulosa cells are encapsulated within a basement membrane, which by the later stages will also have a theca cell layer on the outside; (3) Preantral follicle - an oocyte surrounded by multiple layers of granulosa cells and fluid-filled spaces have started to form in between the granulosa cells, but a fully defined follicular antrum is not present. A theca cell layer is present; (4) Antral follicle - has a fully formed fluid-filled follicular antrum. Oocyte diameter is typically 75 to 100 μ m and is surrounded by a corona radiata and cumulus oophorus. Two or three layers of theca cells are present in small antral follicles, with large antral follicles containing more layers. Antral follicles <2 mm and >2 mm in diameter were quantified separately. Corpora lutea, atretic follicles at any stage and remnant zona pellucida were also quantified. Morphologically, corpora lutea are large structures filled with cells, and can have visible lipid deposits. Atretic follicles were characterized by a degenerating oocyte together with zona pellucida, or remnants of the zona pellucida were the oocyte was no longer present. The granulosa cell layers become disorganized, and depending on the follicle stage when atresia began, a shrinking/collapsing follicular antrum may also be visible, with granulosa cells present within the antrum. Ovarian cysts were classified based on the following criteria ²⁹: a fluid-filled structure either on, in or next to the ovary, and greater than 3.5 mm in diameter. Most cysts likely originated in the rete ovarii ⁴⁰, and were characterized as a fluid-filled structure lined by a layer of cells (flattened, cuboidal or columnar, with or without cilia), and held in tact by a narrow layer of connective tissue. A cystic corpus luteum in one cat was characterized by a fluid-filled structure surrounded by luteinized cells.

Reviewer #4 (Remarks to the Author):
